



# Massive-Parallel Trajectory Calculations version 2.2 (MPTRAC-2.2): Lagrangian transport simulations on Graphics Processing Units (GPUs)

Lars Hoffmann[1], Paul F. Baumeister[1], Zhongyin Cai[1], Jan Clemens[1,2], Sabine Grießbach[1],
Gebhard Günther[2], Yi Heng[3], Mingzhao Liu[1], Kaveh Haghigi Mood[1], Olaf Stein[1], Nicole Thomas[2],
Bärbel Vogel[2], Xue Wu[4,5], and Ling Zou[1]

[1]Jülich Supercomputing Centre, Forschungszentrum Jülich, Jülich, Germany
[2]Institut für Energie- und Klimaforschung (IEK-7), Forschungszentrum Jülich, Jülich, Germany
[3]School of Computer Science and Engineering, Sun Yat-Sen University, Guangzhou, China
[4]Key Laboratory of Middle Atmosphere and Global Environment Observation, Institute of Atmospheric Physics, Chinese Academy of Sciences, Beijing, China
[5]University of Chinese Academy of Sciences, Beijing, China

**Correspondence:** Lars Hoffmann (l.hoffmann@fz-juelich.de)

**Abstract.** Lagrangian models are fundamental tools to study atmospheric transport processes and for practical applications such as dispersion modeling for anthropogenic and natural emission sources. However, conducting large-scale Lagrangian transport simulations with millions of air parcels or more can become numerically rather costly. In this study, we assessed the potential of exploiting graphics processing units (GPUs) to accelerate Lagrangian transport simulations. We ported the

Massive-Parallel Trajectory Calculations (MPTRAC) model to GPUs using the open accelerator (OpenACC) programming model. The trajectory calculations conducted within the MPTRAC model were fully ported to GPUs, i. e., except for feeding in the meteorological input data and for extracting the particle output data, the code operates entirely on the GPU devices without frequent data transfers between CPU and GPU memory. Model verification, performance analyses, and scaling tests of the MPI/OpenMP/OpenACC hybrid parallelization of MPTRAC were conducted on the JUWELS Booster supercomputer

operated by the Jülich Supercomputing Centre, Germany. The JUWELS Booster comprises 3744 NVIDIA A100 Tensor Core GPUs, providing a peak performance of 71.0 PFlop/s. As of June 2021, it is the most powerful supercomputer in Europe and listed among the most energy-efficient systems internationally. For large-scale simulations comprising $10^8$ particles driven by the European Centre for Medium-Range Weather Forecasts' ERA5 reanalysis, the performance evaluation showed a maximum speedup of a factor of 16 due to the utilization of GPUs compared to CPU-only runs on the JUWELS Booster. In the large-scale

GPU run, about 67 % of the runtime is spent on the physics calculations, conducted on the GPUs. Another 15 % of the runtime is required for file-I/O, mostly to read the large ERA5 data set from disk. Meteorological data preprocessing on the CPUs also requires about 15 % of the runtime. Although this study identified potential for further improvements of the GPU code, we consider the MPTRAC model ready for production runs on the JUWELS Booster in its present form. The GPU code provides a much faster time to solution than the CPU code, which is particularly relevant for near-real-time applications of a Lagrangian

transport model.



# 1 Introduction

Lagrangian transport models are frequently applied to study chemical and dynamical processes of the Earth's atmosphere. They have important practical applications in modeling and assessing the dispersion of anthropogenic and natural emissions from local to global scale, for instance, for air pollution (Hirdman et al., 2010; Lee et al., 2014), nuclear accidents (Becker et al., 2007;

Draxler et al., 2015), volcanic eruptions (Prata et al., 2007; D'Amours et al., 2010; Stohl et al., 2011), or wildfires (Forster et al., 2001; Damoah et al., 2004). Concerning studies of atmospheric dynamics of the free troposphere and stratosphere, Lagrangian transport models have been used, for instance, to assess the circulation of the troposphere and stratosphere (Bowman and Carrie, 2002; Konopka et al., 2015; Ploeger et al., 2021), stratosphere-troposphere exchange (Wernli and Bourqui, 2002; James et al., 2003; Stohl et al., 2003), mixing of air (Konopka et al., 2005, 2007; Vogel et al., 2011), and large-scale features of the

atmospheric circulation, such as the polar vortex (Grooß et al., 2005; Wohltmann et al., 2010) or the Asian monsoon anticyclone (Bergman et al., 2013; Vogel et al., 2019; Legras and Bucci, 2020).

A wide range of Lagrangian transport models has been developed for research studies and operational applications during the past decades (Draxler and Hess, 1998; McKenna et al., 2002b,a; Lin et al., 2003; Stohl et al., 2005; Jones et al., 2007; Stein et al., 2015; Sprenger and Wernli, 2015; Pisso et al., 2019). While Eulerian models represent fluid flows in the atmo-

sphere based on the flow between regular grid boxes of the model, Lagrangian models represent the transport of trace gases and aerosols based on large sets of air parcel trajectories following the fluid flow. Both approaches have distinct advantages and disadvantages. Lagrangian models are particularly suited to study fine-scale structures, filamentary transport, and mixing processes in the atmosphere, as their spatial resolution is not inherently limited to the resolution of Eulerian grid boxes and numerical diffusion is particularly low for these models. However, Lagrangian transport simulations may become rather costly

because subgrid-scale processes, such as diffusion and mesoscale wind fluctuations, need to be represented in a statistical sense, i. e., by adding stochastic perturbations to large sets of air parcel trajectories.

In this study, the Massive-Parallel Trajectory Calculations (MPTRAC) model is applied to exploit the potential of conducting Lagrangian transport simulations on graphics processing units (GPUs). MPTRAC was first described by Hoffmann et al. (2016), discussing Lagrangian transport simulations for volcanic eruptions with different meteorological data sets. Heng et al.

(2016) and Liu et al. (2020) discussed inverse transport modeling to estimate volcanic sulfur dioxide emissions from the Nabro eruption in 2011 based on large-scale simulations with MPTRAC. Wu et al. (2017, 2018) conducted case studies on global transport of volcanic sulfur dioxide emissions of Sarychev Peak in 2009 and Mount Merapi in 2010. Cai et al. (2021) studied the Raikoke volcanic eruption in June 2019. Zhang et al. (2020) applied MPTRAC to study aerosol variations in the upper troposphere and lower stratosphere over the Tibetan Plateau. Smoydzin and Hoor (2021) used the model to assess

the contribution of Asian emissions to upper tropospheric carbon monoxide over the remote Pacific. Hoffmann et al. (2017) presented an intercomparison of meteorological analyses and trajectories in the Antarctic lower stratosphere with Concordiasi superpressure balloon observations. Rößler et al. (2018) investigated the accuracy and computational efficiency of different integration schemes to solve the trajectory equation with MPTRAC.





The idea of using specialized computation units for scientific computing goes back to the early 1980s when co-processors like Intel's 8087 and 8231 were introduced. It is more than 20 years since graphics processing units (GPUs) were leveraged for non-graphical general-purpose calculations (Hoff et al., 1999) and a decade since the first GPU-accelerated cluster appeared on the top 500 list of supercomputers (Dongarra et al., 2021). Today, six of the ten fastest supercomputers on the top500 list and nine out of ten on the green500 list are GPU-accelerated machines. Most of the pre-exascale systems (CINECA, 2020; CSC, 2020) and likely the first exascale machine will be accelerated by GPUs (DOE, 2019a,b; LLNL, 2020). Besides energy efficiency, the software and hardware maturity contributed to the popularity of GPUs. GPUs are not only interesting high performance computing (HPC) research topics anymore but essential workhorses for modern scientific computing.

The aim of the present study is to investigate the potential of using GPUs for accelerating Lagrangian transport simulations, a topic first explored by Molnár et al. (2010). Large-scale and long-term Lagrangian transport simulations for climate studies or inverse modeling applications can become very compute-intensive (Heng et al., 2016; Liu et al., 2020). GPUs bear the potential to not only calculate the solutions of Lagrangian transport problems more quickly, but also to obtain them in a much more energy-efficient manner. For this study, we ported our existing Lagrangian transport model MPTRAC to GPUs by means of the open accelerators (OpenACC) programming model. Next to offloading calculations to GPUs, the code is also capable of distributing computing tasks employing the Message Passing Interface (MPI) over the compute nodes and the Open Multi-Processing (OpenMP) over the CPU cores of a heterogeneous supercomputer. A detailed evaluation and performance assessment of MPTRAC on GPUs was conducted on the Jülich Wizard for European Leadership Science (JUWELS) system (Jülich Supercomputing Centre, 2019) at the Jülich Supercomputing Centre, Germany.

Lagrangian transport simulations are often driven by global meteorological reanalyses or forecast data sets. The MPTRAC model has been used with the National Centers for Environmental Prediction and National Center for Atmospheric Research (NCEP/NCAR) Reanalysis 1 (Kalnay et al., 1996), the National Aeronautics and Space Administration (NASA) Modern-Era Retrospective analysis for Research and Applications (Rienecker et al., 2011; Gelaro et al., 2017) as well as the European Centre for Medium-Range Weather Forecasts (ECMWF) ERA-Interim reanalysis (Dee et al., 2011) and operational analyses. More recently, we also applied ECMWF's fifth generation reanalysis, ERA5 (Hersbach et al., 2020), to drive simulations with the MPTRAC model. Providing hourly data at 31 km horizontal resolution on 137 vertical levels from the surface up to 0.01 hPa (about 80 km of height) on global scale, ERA5 poses a particular challenge for Lagrangian transport modeling due to the large amount of data that needs to be handled (Hoffmann et al., 2019). In this study, we conducted the model verification and performance analyses of MPTRAC with ERA5 data, as this is the data set we intend to mainly use in future works, to benefit from high resolution and other improvements of the forecasting model and data assimilation scheme. It also needs to be taken into account that the production of ERA-Interim stopped in August 2019 in favor of ERA5.

We provide a comprehensive description of the MPTRAC model in Sect. 2 of this paper. Next to describing the different implemented algorithms and the requirements and options for model input and output data, we discuss the parallelization strategy and the porting of the code to GPUs in more detail. Model verification, as well as performance and scaling analyses are discussed in Sect. 3. As the CPU code of MPTRAC was already verified in earlier studies, we mainly focus on the direct comparisons of CPU and GPU calculations in the present work. The model evaluation covers comparisons of both, individual kinematic





trajectory calculations and global simulations of synthetic tracer distributions, including the effects of diffusion, convection,
and chemical lifetime. We assessed the performance and scaling of all three components of the MPI/OpenMP/OpenACC hybrid
parallelization on a single device and in a multi-GPU setup. Section 4 provides the summary and conclusions of this study.

## 2 Model description

### 2.1 Overview on model features and code structure

Figure 1 provides a simplified overview on the code structure and most of the features of the MPTRAC model. The call graph
shown here was derived automatically from the C code of MPTRAC by means of the `cflow` tool, but it has been edited and
strongly simplified in order to present only the most relevant features. Following the input-process-output (IPO) pattern, the
call graph reveals the principle input functions (`read_*`), the processing functions (`module_*`), and the output functions
(`write_*`) of MPTRAC.

Three main input functions are available, i. e., `read_ctl` to read the model control parameters, `read_atm` to read
the initial particle data, and `read_met` to read the meteorological input data. The `read_met` function further splits into
functions that deal directly with reading of the meteorological data files (`read_met_grid`, `read_met_levels`, and
`read_met_surface`), extrapolation and sampling of the meteorological data (`read_met_extrapolate`, `read_met_`
`periodic`, `read_met_sample`, and `read_met_detrend`), and the calculation of additional meteorological variables
(`read_met_geopot`, `read_met_pv`, `read_met_pbl`, `read_met_tropo`, `read_met_cloud`, and `read_met_`
`cape`), also referred to as meteorological data preprocessing in this work. Although the additional meteorological variables
are often available directly via the meteorological input data files, the MPTRAC model allows users to recalculate these data
consistently from different meteorological input data sets. The MPTRAC model input data are further discussed in Sect. 2.2.

The processing functions of MPTRAC provide the capabilities to calculate kinematic trajectories of the particles using
given $\omega$-velocities (`module_advection`) and to add stochastic perturbations to the trajectories to simulate the effects of
diffusion and subgrid-scale wind fluctuations (`module_diffusion_turb` and `module_diffusion_meso`). The func-
tions `module_convection` and `module_ sedi` may alter the particle positions along the trajectories by simulating the
effects of unresolved convection and sedimentation. The function `module_isosurf` allows us to vertically constrain the
particle positions to different types of isosurfaces. The function `module_position` enforces the particle to remain within
the boundaries of the model domain. The function `module_meteo` allows us to sample the meteorological data along the
trajectories at fixed time intervals. The functions `module_decay`, `module_oh_chem`, `module_dry_deposition`,
`module_wet_deposition`, and `module_bound_cond` affect the mass or volume mixing ratios of the particles based
on a given e-folding lifetime, chemical decomposition by the hydroxyl radical, dry and wet deposition, or by enforcing bound-
ary conditions, respectively. Most of the chemistry and physics modules are presented in more detail in Sect. 2.3.

Finally, the model output is directed via a generic function (`write_output`) towards individual functions that can be used
to write particle data (`write_atm`), gridded data (`write_grid`), or ensemble data for groups of particles (`write_ens`).
The functions `write_csi`, `write_prof`, `write_sample`, and `write_station` can be used to sample the model data

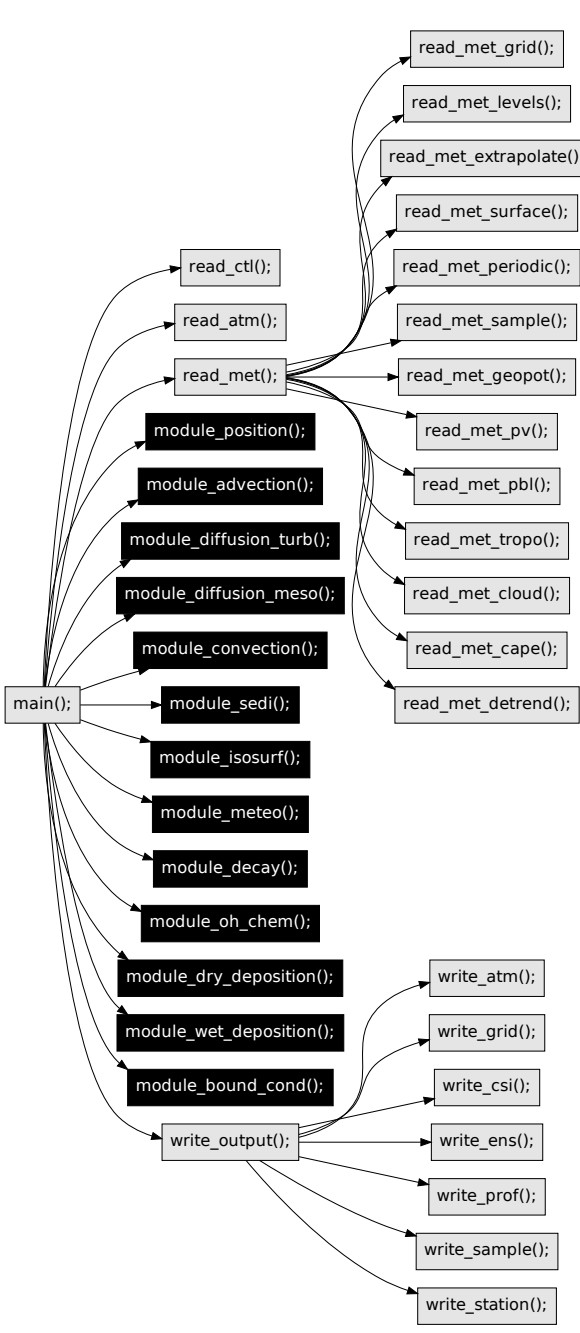

**Figure 1.** Call graph of the most relevant functions of the MPTRAC model. The individual functions are sorted following the IPO approach, see text for details. Black boxes highlight parts of the code that are ported to GPUs.





in specific manners to enable comparisons with observational data, such as satellite measurements or data from measurement stations. The model output is further discussed in Sect. 2.4.

All the processing functions (`module_*`) are highlighted in Fig. 1 to indicate that they have been ported to GPUs in the present work. Porting these processing functions to GPUs is the natural choice as these functions typically require most of the computing time, in particular for simulations with many particles. The input and output functions cannot easily be ported to GPUs as the input and output operations are directed over the CPUs and utilize CPU memory in the first place. The approach used for porting the physics and chemistry modules to GPUs and the parallelization strategy of MPTRAC are discussed in more detail in Sect. 2.5.

## 2.2 Model input data

### 2.2.1 Initial air parcel positions

In order to enable trajectory calculations, MPTRAC requires an input file providing the initial positions of the air parcels. The initial positions are defined in terms of time, log-pressure height, longitude, and latitude. Internally, MPTRAC applies pressure as its vertical coordinate. For convenience, the initial pressure $p$ is specified in terms of a log-pressure height $z$ obtained from the barometric formula, $p(z) = p_0 \exp\left(-\frac{z}{H}\right)$, with a fixed reference pressure $p_0 = 1013.25\,\mathrm{hPa}$ and scale height $H = 7\,\mathrm{km}$. Longitude and latitude refer to a spherical Earth approximation, with a constant radius of curvature of $R_e = 6367.421\,\mathrm{km}$.

Internally, MPTRAC uses the time in seconds since 2000-01-01, 00:00 UTC as its time coordinate. Tools are provided along with the model to convert the internal time from and to UTC time format. The internal time step $\Delta t$ of the model simulations can be adjusted via a control flag. MPTRAC allows for arbitrary start and end times of the trajectories by adjusting (shortening) the initial and the final time steps to synchronize with the internal model time steps $\Delta t$. A control flag is provided to specify whether the model should conduct the trajectory calculations forward or backward in time.

Unless the initial positions of the trajectories are derived from measurements, a set of tools provided with MPTRAC can be used to create initial air parcel files. The tool `atm_init` creates a regular grid of air parcel positions in time, log-pressure height, longitude, and latitude. Around each grid point and time, the tool can create random distributions of multiple particles by applying either uniform distributions with given widths or normal distributions with given standard deviations. The total mass of all air parcels can be specified, which is distributed equally over all air parcels. Alternatively, a fixed volume mixing ratio for each air parcel can be specified. With this tool, initializations for a wide range of sources (e. g., point source, vertical column, surface, or volume source) with spontaneous or continuous emissions can be created.

A tool to modify air parcel positions is `atm_split`. It can be used to create a larger set of air parcel positions based on randomized resampling from a given, smaller set of air parcel positions. It splits the initial small set into a larger set of air parcels. The total number of air parcels that is desired for the large set can be specified, which is helpful because it provides control over the computational time needed for a simulation. The probability by which an air parcel is selected during the resampling can be weighted by its mass. The total mass of the initial air parcels can be retained or adjusted during the resampling process. During the resampling process, uniform or Gaussian stochastic perturbations in space and time can be





applied to the air parcel positions. These stochastic perturbations can be used to spread the air parcels over a larger volume, which might represent the measurement volume covered by an instrument, for example. Finally, a vertical weighting function can be applied in the resampling process in order to redistribute the air parcel positions to follow the vertical sensitivity of a satellite sounder, for instance.

### 2.2.2 Requirements and preprocessing of meteorological input data

Next to the particle positions, the other main input data required by MPTRAC are meteorological data from a global forecasting or reanalysis system. At a minimum, MPTRAC requires 3-D fields of temperature $T$, zonal wind $u$, and meridional wind $v$. The vertical velocity $\omega$, specific humidity $q$, ozone mass mixing ratio $x_{O3}$, cloud liquid water content (CLWC), cloud rain water content (CRWC), cloud ice water content (CIWC), and cloud snow water content (CSWC) are optional, but often needed for specific applications. The model also requires the surface pressure $p_s$ or log-surface pressure $(\ln p_s)$ and the geopotential height

at the surface $Z_g$ to be provided as 2-D fields. Optionally, near surface temperature $T_{2m}$ and near surface winds $(u_{10m}, v_{10m})$ can be ingested from the input files.

    The model requires the input data in terms of separate netCDF files at regular time steps. The meteorological data have to be provided on a regular horizontal latitude $\times$ longitude grid. The model internally uses pressure $p$ as its vertical coordinate, i. e., the meteorological data should be provided on pressure levels. However, the model can also ingest meteorological data on

other types of vertical levels, e. g., hybrid sigma vertical coordinates $\eta$ as used by ECMWF's Integrated Forecast System (IFS). In this case, the model requires $p$ as a 3-D coordinate on the input model levels in order to facilitate vertical interpolation of the data to a given list of pressure levels for further use in the model.

    Once the mandatory and optional meteorological input variables have been read in from disk, MPTRAC provides the capability to calculate additional meteorological variables from the input data. This includes the options to calculate geopotential

heights, potential vorticity, the tropopause height, cloud properties such as cloud layer depth and total column cloud water, the convective available potential energy (CAPE), and the planetary boundary layer (PBL). Having the option to calculate these additional meteorological data directly in the model helps to reduce the disk space needed to safe them as input data. This is particularly relevant for large input data sets such as ERA5. It is also an advantage in terms of consistency, if the same algorithms can be applied to infer the additional meteorological variables from different meteorological data sets. The algorithms

and selected examples on the meteorological data preprocessing are presented in the electronic supplement of this paper.

### 2.2.3 Boundary conditions for meteorological data

    The upper boundary of the model is defined by the lowest pressure level of the meteorological input data. The lower boundary of the model is defined by the 2-D surface pressure field $p_s$ at a given time. If an air parcel leaves the pressure range covered by the model during the trajectory calculations, it will be pushed back to the lower or upper boundary of the model, respectively.

Therefore, the particles will follow the horizontal wind and slide along the surface or the upper boundary until they encounter an up- or downdraft, respectively. For the surface, this approach follows Draxler and Hess (1997). We recognize that pressure might not be the best choice for the vertical coordinate of a Lagrangian model, in particular for the boundary layer, because





a broad range of surface pressure variations needs to be represented appropriately by a fixed set of pressure levels. However, as the scope of MPTRAC is on applications covering the free troposphere and the stratosphere, we consider the limitation of

having reduced vertical resolution in terms of pressure levels near the surface acceptable for now.

For global meteorological data sets, MPTRAC applies periodic boundary conditions in longitude. During the trajectory calculations, air parcels may sometimes cross either the pole or the longitudinal boundary of the meteorological data. If an air parcel crosses the North or South Pole, its longitude will be shifted by $180°$ and the latitude will be recalculated to match the $\pm 90°$ latitude range, accordingly. If an air parcel leaves the longitude range, it will be re-positioned by a shift of $360°$ to

fit the range again. The model works with both, meteorological data provided at $-180° \cdots +180°$ or $0° \ldots 360°$ of longitude. Following Stohl et al. (2005), we mirror data from the western boundary at the eastern boundary of the model, in order to achieve full coverage of the $360°$ longitude range and to avoid extrapolation of the meteorological input data.

### 2.2.4  Interpolation and sampling of meteorological data

In MPTRAC, 4-D linear interpolation is applied to sample the meteorological data at any given position and time. Although

higher-order interpolation schemes may achieve better accuracy, linear interpolation is considered a standard choice in Lagrangian particle dispersion models (Bowman et al., 2013). The approach requires that two time steps of the meteorological input data are kept in memory at each time. Higher-order interpolation schemes in time would require more time steps of the input data and would therefore require more memory.

Following the memory layout of the data structures used for the meteorological input data in our code and to make efficient

use of memory caches, the linear interpolations are conducted first in the vertical domain, followed by latitude and longitude, and finally in time. Interpolations can be conducted for a single variable or all the meteorological data at once. Two separate index look-up functions are implemented for regularly gridded data (longitude and latitude) and for irregularly gridded data (pressure). Interpolation weights are kept in caches to improve efficiency of the calculations.

For various studies, it is interesting to investigate the sensitivity of the results with respect to the spatial and temporal

resolution of the meteorological input data. For instance, Hoffmann et al. (2019) assessed the sensitivity of Lagrangian transport simulations for the free troposphere and stratosphere with respect to downsampled versions of the ERA5 reanalysis data set. To enable such studies, we implemented an option for spatial downsampling of the meteorological input data. The downsampling is conducted in two steps. In the first step, the full resolution data are smoothed by applying triangular filters in the respective dimensions. Smoothing of the data is required in order to avoid aliasing effects. In the second step, the data are downsampled

by selecting only each $(l, m, n)$-th grid point in longitude, latitude, and pressure, respectively.

Note that downsampling has not been implemented rigorously for the time domain. Nevertheless, the user can specify the time interval at which the meteorological input data should be ingested. If filtering of the downsampled data is required, downsampling in time needs to be done externally by the user by applying tools such as the Climate Data Operators (CDO) (Schulzweida, 2014) to preprocess the input data of different time steps accordingly. In any case, it needs to be carefully

considered whether the meteorological input data represent instantaneous or time-averaged diagnostics at the given time steps of the data.





### 2.3 Chemistry and physics modules

#### 2.3.1 Advection

The advection of an air parcel, i.e., the position $\boldsymbol{x}(t)$ at time $t$ for a given wind and velocity field $\boldsymbol{v}(\boldsymbol{x},t)$, is given by the

trajectory equation,

$$\frac{d\boldsymbol{x}}{dt} = \boldsymbol{v}(\boldsymbol{x},t). \tag{1}$$

Here, the position $\boldsymbol{x}$ is provided in a meteorological coordinate system, $\boldsymbol{x} = (\lambda, \phi, p)$, with longitude $\lambda$, latitude $\phi$, and pressure $p$. The velocity vector $\boldsymbol{v} = (u, v, \omega)$ is composed of the zonal wind component $u$, the meridional wind component $v$, and the vertical velocity $\omega$, respectively. Based on its accuracy and computational efficiency for trajectory calculations (Rößler et al.,

2018), we apply the explicit midpoint method to solve the trajectory equation,

$$\boldsymbol{x}(t+\Delta t) = \boldsymbol{x}(t) + \Delta t\, \boldsymbol{v}\left\{ \boldsymbol{x}(t) + \frac{\Delta t}{2} \boldsymbol{v}\left[\boldsymbol{x}(t), t\right], t + \frac{\Delta t}{2}\right\}. \tag{2}$$

The model time step $\Delta t$ controls the trade-off between the accuracy of the solution and the computational effort. Similar to the Courant–Friedrichs–Lewy (CFL) condition, $\Delta t$ should, in general, be selected such that the steps $\Delta \boldsymbol{x} = \boldsymbol{x}(t+\Delta t) - \boldsymbol{x}(t)$ remain smaller than the grid box size of the meteorological data. MPTRAC will issue a warning, if $\Delta t > \Delta \lambda_{met} R_e / u_{max}$

for a longitudinal grid spacing $\Delta \lambda_{met}$ of the meteorological data, the mean radius of Earth $R_e$, and a maximum zonal wind speed, $u_{max} = 150\,\mathrm{m\,s^{-1}}$. A default time step $\Delta t = 180\,\mathrm{s}$ has been selected to match the effective horizontal resolution of $\Delta x \approx 31\,\mathrm{km}$ of ECMWF's ERA5 reanalysis.

Calculations of the numerical solution of the trajectory equation require coordinate transformations between Cartesian coordinate distances $(\Delta x, \Delta y, \Delta z)$ and spherical coordinate distances $(\Delta \lambda, \Delta \phi, \Delta p)$,

$$\Delta \lambda = \frac{\Delta x}{R_e \cos(\phi)}, \tag{3}$$

$$\Delta \phi = \frac{\Delta y}{R_e}, \tag{4}$$

$$\Delta p = -\frac{p}{H_0} \Delta z. \tag{5}$$

The vertical coordinate transformation between $\Delta p$ and $\Delta z$ uses an average scale height of $H_0 = 7\,\mathrm{km}$. However, note that the vertical transformation is currently not needed to solve Eq. (2), because the vertical velocity is already provided in terms of

pressure change, $\omega = dt/dp$, in the meteorological data. The transformation is reported here for completeness, as it is required in other parts of the model. For convenience, we assume $\lambda$ and $\phi$ to be given in radians, i.e., scaling factors of $\pi/180°$ and $180°/\pi$ to convert from degrees to radians and vice versa do not need to be introduced here.

The coordinate transformation between $\Delta \lambda$ and $\Delta x$ in Eq. (3) bears numerical problems due to the singularities near the poles. Although this problem might be solved by means of a change to a more suitable coordinate system at high latitudes, we

implemented a rather simple solution. Near the poles, for $|\lambda| > \lambda_{max}$, we discard the longitudinal distance $\Delta \lambda$ calculated from



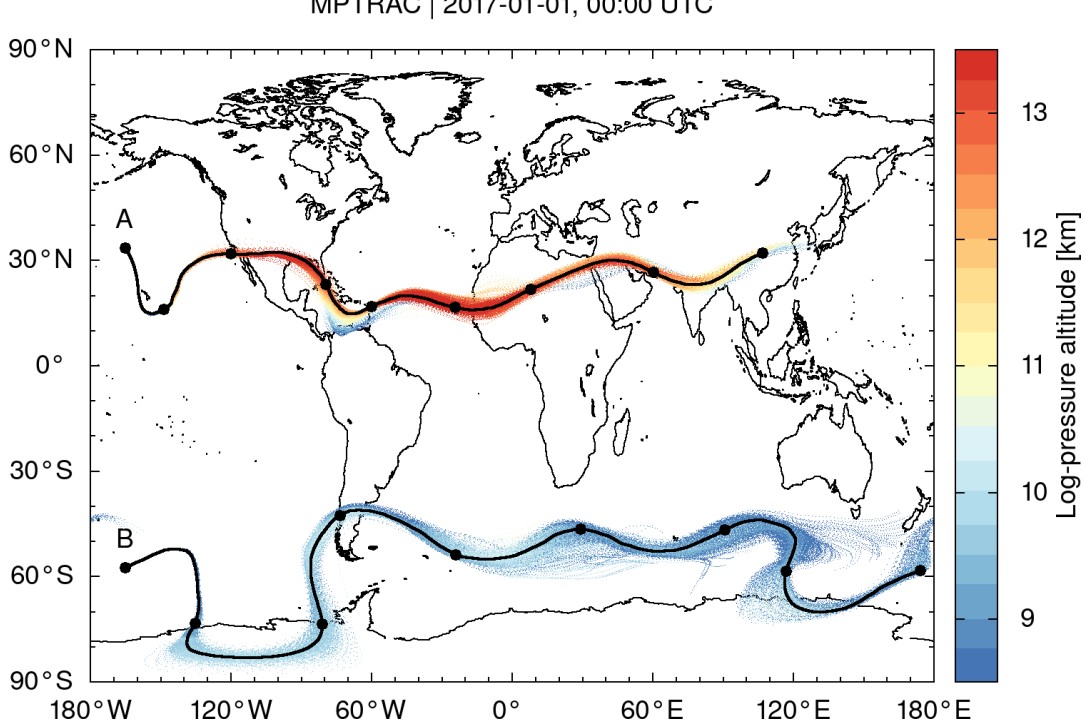

**Figure 2.** Trajectory calculations for the Northern Hemisphere subtropical jet (case A) and the Southern Hemisphere polar jet (case B) from 1 January 2017, 00:00 UTC to 9 January 2017, 00:00 UTC. Trajectories were launched at positions A and B at log-pressure altitudes of 11.25 and 8.5 km, respectively. Calculations used hourly ERA5 horizontal winds and vertical velocity data for input. Black curves show trajectories calculated without diffusion. Black dots indicate 24 h time intervals. Colored dots show calculations for two sets of 1000 trajectories for cases A and B with turbulent diffusion and subgrid-scale wind fluctuations being considered.

Eq. (3) and use $\Delta\lambda = 0$ instead. The high latitude threshold was set to $\lambda_{max} = 89.999°$, a distance of about 110 m from the poles, following an assessment of its effect on trajectory calculations conducted by Rößler (2015).

As an example, Fig. 2 shows the results of kinematic trajectory calculations for particles located in the vicinity of the Northern Hemisphere subtropical jet (case A) and the Southern Hemisphere polar jet (case B) in January 2017 using ERA5
data for input. The trajectories were launched at (33.5°N, 165°W) at 11.25 km log-pressure altitude for the subtropical jet and at (57.5°S, 165°W) at 8.5 km of log-pressure altitude for the polar jet on 1 January 2017, 00:00 UTC, respectively. The trajectory seeds are located close to the horizontal wind speed maxima at 165°W in the upper troposphere and lower stratosphere at the given time. In both cases, the trajectories nearly circumvent the Earth during the time period of 8 days covered by the trajectory calculations. Significant horizontal meandering and height oscillations with the jet streams are revealed. The trajectories were
calculated with the explicit midpoint method and a default integration time step of 180 s. Varying the time step between 45, 90, 180, 360, and 720 s did not reveal any significant deviations of the results (not shown).





### 2.3.2 Turbulent diffusion

Rather complex parametrizations of atmospheric diffusivity are available for the planetary boundary layer (e. g., Draxler and Hess, 1997; Stohl et al., 2005), which have not been implemented in MPTRAC, so far. Much less is known about the dif-
fusivities in the free troposphere and in the stratosphere, which are in the scope of our model. In MPTRAC, the effects of atmospheric diffusion are simulated by adding stochastic perturbations to the positions $x$ of the air parcels at each time step $\Delta t$ of the model,

$$\Delta x_i(t + \Delta t) = x_i(t) + \sqrt{2 D_i \Delta t}\xi_i. \tag{6}$$

This method requires a vector $\boldsymbol{\xi} = (\xi_x, \xi_y, \xi_z)$ of random variates to be drawn from the standard normal distribution for each
air parcel at each time step. The vector of diffusivities $\boldsymbol{D} = (D_x, D_x, D_z)$ is composed of the horizontal diffusivity $D_x$ and the vertical diffusivity $D_z$. The model allows specifying $D_x$ and $D_z$ separately for the troposphere and stratosphere as control parameters. Following choices made for the FLEXPART model (Stohl et al., 2005), default values of $D_x = 50\,\mathrm{m}^2\,\mathrm{s}^{-1}$ and $D_z = 0$ are selected for the troposphere and $D_x = 0$ and $D_z = 0.1\,\mathrm{m}^2\,\mathrm{s}^{-1}$ are selected for the stratosphere. Diffusivities will therefore change, if an air parcel intersects the tropopause. A smooth transition between tropospheric and stratospheric diffusivities is
created by linear interpolation of $D_x$ and $D_z$ within a $\pm 1\,\mathrm{km}$ log-pressure altitude range around the tropopause.

### 2.3.3 Subgrid-scale wind fluctuations

In addition to turbulent diffusion, the effects of unresolved subgrid-scale winds, also referred to as mesoscale wind perturbations, are considered. The starting point for this approach is the separation of the horizontal wind and vertical velocity vector $\boldsymbol{v}$ into the grid-scale mean $\bar{\boldsymbol{v}}$ and the subgrid-scale perturbations $\boldsymbol{v}'$,

$$\boldsymbol{v} = \bar{\boldsymbol{v}} + \boldsymbol{v}'. \tag{7}$$

It is further assumed that the mean wind $\bar{\boldsymbol{v}}$ is given by the coarse-grid meteorological data, whereas the wind perturbations $\boldsymbol{v}'$ need to be parametrized. The subgrid-scale wind perturbations are calculated by means of the Langevin equation,

$$v_i'(t + \Delta t) = r v_i'(t) + \sqrt{1 - r^2}(f\sigma_i)^2 \xi_i, \tag{8}$$

$$r = 1 - 2\frac{\Delta t}{\Delta t_{met}}. \tag{9}$$

From a mathematical point of view, this is a Markov chain or a random walk, which adds temporally correlated stochastic perturbations to the winds over time. The degree of correlation depends on the correlation coefficient $r$, and therefore on the time step $\Delta t$ of the model and the time step $\Delta t_{met}$ of the meteorological data. The variance $(f\sigma_i)^2$ of the random component added at each time step depends on the grid-scale variance $\sigma_i^2$ of the horizontal wind and vertical velocity data and a scaling factor $f$, which is used for downscaling to the subgrid scale. The scaling factor $f$ needs to be specified as a control parameter of
the model. The default value is $f = 40\,\%$, following a choice made for the FLEXPART model (Stohl et al., 2005). For each air parcel, the grid-scale variance $\sigma_i^2$ is calculated from the neighboring eight grid boxes and two time steps of the meteorological





data. To make computations more efficient, the grid-scale variance of each parcel is kept in a cache. As before, $\boldsymbol{\xi}$ is a vector of random variates to be drawn from the standard normal distribution.

Figure 3 shows zonal means of the grid-scale standard deviations of the horizontal wind and the vertical velocity of ERA5
data for 1 January 2017. Although they are treated separately in the parametrization, we here combined the zonal and meridional wind components for convenience, noting that the standard deviations of the horizontal wind are largely dominated by the zonal wind component. The grid-scale standard deviations of the horizontal wind become as large $20\,\mathrm{m\,s^{-1}}$ in the tropical mesosphere. The vertical velocity standard deviations are largest (up to $0.08\,\mathrm{m\,s^{-1}}$) at the Northern Hemisphere mid-latitude stratopause. Standard deviations are generally much lower in the troposphere and the lower stratosphere. However, some
pronounced features of the horizontal wind (up to $3\,\mathrm{m\,s^{-1}}$) are found at the tropopause in the tropics and at northern and southern mid-latitudes and for the vertical velocities (up to $0.02\,\mathrm{m\,s^{-1}}$) in the tropical and mid-latitude upper troposphere. The parametrized subgrid-scale wind fluctuations vary significantly with time and location and the atmospheric background conditions.

Figure 2 illustrates the effects of the simulated turbulent diffusion and the subgrid-scale winds on the trajectories calculated
for the jet streams. Sets of 1000 particles were released at the same starting point for both cases. The trajectory sets reveal significant horizontal and vertical spread over time. The spread mostly increases in regions where the jets are meandering and where notable horizontal and vertical wind shears are present. We calculated absolute horizontal and vertical transport deviations (AHTDs and AVTDs, see Sect. 2.4.3) in order to quantify the effects of simulated diffusion and subgrid-scale wind fluctuations on the jet stream trajectory calculations. Results for case A, the Northern Hemisphere subtropical jet, are shown
in Fig. 4. The analysis of the transport deviations showed that the subgrid-scale wind fluctuations were the major source of the spread of the particles. After 8 days of trajectory time, the AHTDs and AVTDs due to the subgrid-scale winds alone (i. e., $D_x = D_z = 0$ and $f = 40\,\%$) were about $550\,\mathrm{km}$ and $450\,\mathrm{m}$, respectively. In contrast, the corresponding transport deviations due to the constant diffusivities only (i. e., $f = 0$, $D_x = 50\,\mathrm{m^2\,s^{-1}}$ in the troposphere, and $D_z = 0.1\,\mathrm{m^2\,s^{-1}}$ in the stratosphere) were about $150\,\mathrm{km}$ for the AHTDs and $200\,\mathrm{m}$ for the AVTDs.

Note that several studies with Lagrangian models provided estimates of atmospheric diffusivities (Legras et al., 2003, 2005; Pisso et al., 2009), but the data required for the free troposphere and stratosphere for the parametrizations are typically still not well constrained. In addition, in earlier work we found that the parametrizations of diffusion and subgrid-scale winds applied here may yield different particle spread for different meteorological data sets, even if the same parameter choices are applied (Hoffmann et al., 2017). As an example, Fig. 4 shows results of sensitivity tests on the parameter choices for the diffusion
and subgrid-scale wind parametrizations. The parameter choices may need careful tuning for each individual meteorological data set. However, this also opens the possibility for potentially interesting considerations regarding sensitivities to the relative importance of diffusion and sub-grid wind fluctuations.



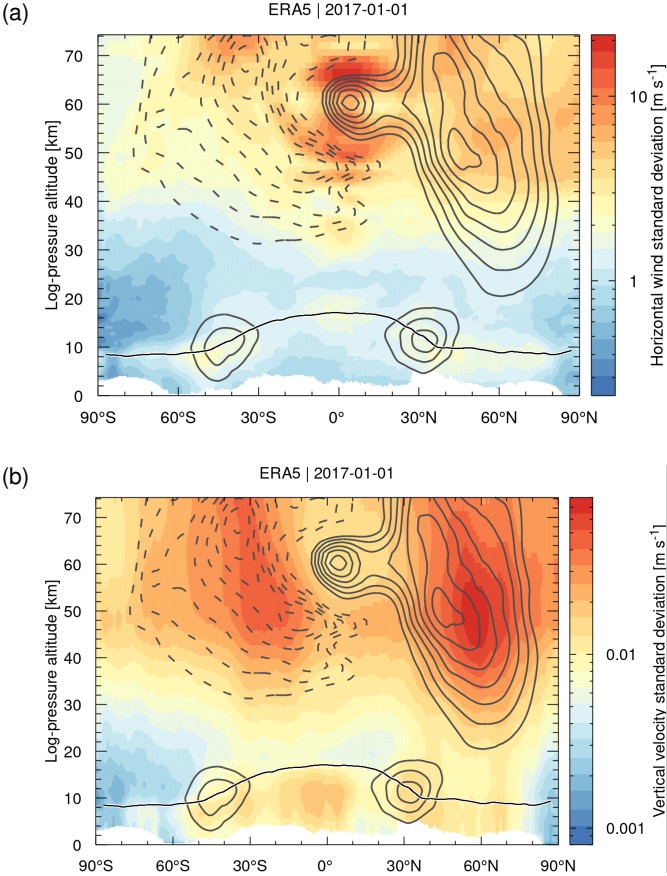

**Figure 3.** The contour surfaces show the zonal mean standard deviations of (a) the grid-scale horizontal wind and (b) the grid-scale vertical velocity on 1 January 2017 as calculated by MPTRAC from ERA5 data. Gray curves show positive (solid) and negative (dashed) zonal mean zonal wind at levels of $\pm 20, 30, 40, \ldots \mathrm{m\,s^{-1}}$, respectively. The black curve shows the zonal mean dynamical tropopause.



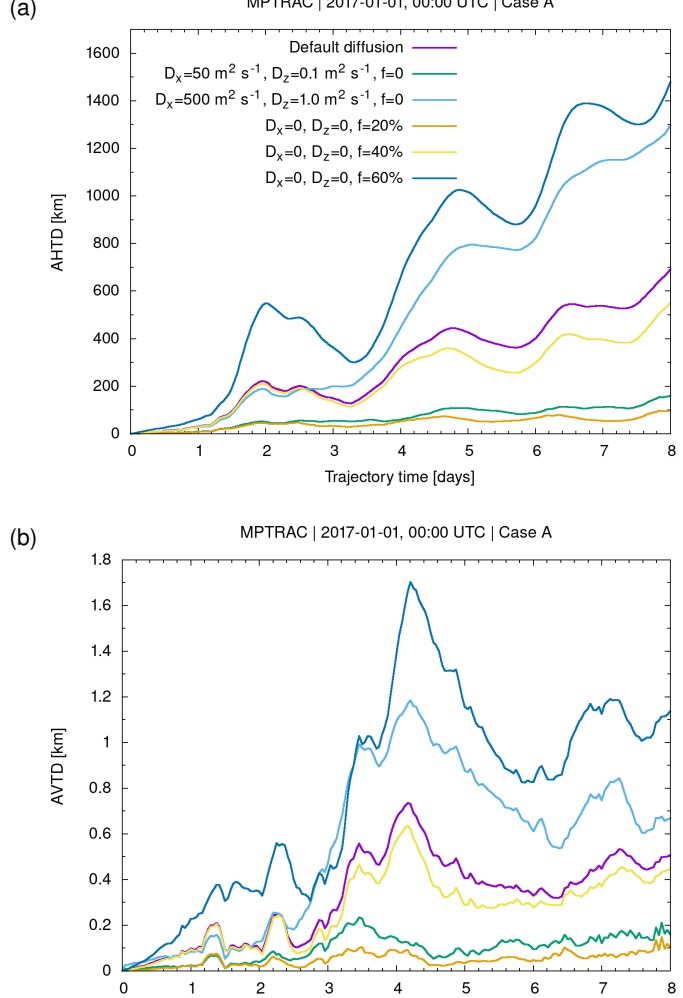

**Figure 4.** Absolute horizontal transport deviations (AHTDs, a) and absolute vertical transport deviations (AVTDs, b) between trajectory calculations with and without simulated diffusion and subgrid-scale winds. Statistics refer to case A presented in Fig. 2. The simulations use different parameter choices of the diffusivities $D_x$ and $D_z$ and the scaling factor $f$ for the subgrid-scale variances (see plot key). Default parameters (purple curve) are $D_x = 50\,\mathrm{m^2\,s^{-1}}$ in the troposphere, $D_z = 0.1\,\mathrm{m^2\,s^{-1}}$ in the stratosphere, and $f = 40\,\%$.





### 2.3.4 Convection

The spatial resolution of global meteorological input data is often too coarse to allow for explicit representation of convective
up- and downdrafts. Although the downdrafts may occur on larger horizontal scales, the updrafts are usually confined to
horizontal scales below a few kilometers. Furthermore, convection occurs on timescales of a few hours, i. e., hourly ERA5 data
may better represent convection than six-hourly ERA-Interim data. A parametrization to better represent unresolved convective
up- and downdrafts in global simulations was implemented in MPTRAC, which is similar to the convection parametrization
implemented in the HYSPLIT and STILT models (Draxler and Hess, 1997; Gerbig et al., 2003). In the present study, the
convection parametrization was applied for synthetic tracer simulations as discussed in Sect. 3.3.

The convection parametrization requires as input the convective available potential energy (CAPE) and the equilibrium
level from the meteorological data. These data need to be interpolated to the horizontal position of each air parcel. If the
interpolated CAPE value is larger than a threshold $CAPE_0$, which is an important control parameter of the parametrization, it
is assumed that the up- and downdrafts within the convective cloud are strong enough to yield a well-mixed vertical column.
The parametrization will randomly redistribute the air parcels in the vertical column between the surface and the equilibrium
level, which is taken as an estimate of the cloud top height. In order to achieve a well-mixed column, the random distribution
of the particles needs to be vertically weighted by air density.

The globally applied threshold $CAPE_0$ can be set to a value near zero, implying that convection will always take place every-
where below the equilibrium level. This approach is referred to as the "extreme convection method". It will provide an upper
limit to the effects of unresolved convection. On the contrary, completely switching off the convection parametrization will
provide a lower limit for the effects of unresolved convection. Intermediate states can be simulated by selecting other specific
values of the threshold $CAPE_0$, which is an important tuning parameter of the parametrization. The electronic supplement of
this paper provides some guidance on how different choices of $CAPE_0$ can affect the simulated convection.

### 2.3.5 Sedimentation

In order to take into account the gravitational settling of particles, the sedimentation velocity $v_s$ needs to be calculated. Once
$v_s$ is known, it can be used to calculate the change of the vertical position of the particles over the model time step $\Delta t$. In
MPTRAC, $v_s$ is calculated for spherical particles following the method described by Jacobson (1999). In the first step, we
calculate the density $\rho$ of dry air,

$$\rho = \frac{p}{RT},\tag{10}$$

from pressure $p$, temperature $T$, and the specific gas constant $R$ of dry air. Next, the dynamic viscosity of air,

$$\eta = 1.8325 \times 10^{-5} \frac{\mathrm{kg}}{\mathrm{m\,s}} \left( \frac{416.16\,\mathrm{K}}{T + 120\,\mathrm{K}} \right) \left( \frac{T}{296.16\,\mathrm{K}} \right)^{1.5},\tag{11}$$

and the thermal velocity of an air molecule,

$$v = \sqrt{8RT},\tag{12}$$





are used to calculate the mean free path of an air molecule,

$$\lambda = \frac{2\,\eta}{\rho\,v}, \tag{13}$$

as well as the Knudsen number for air,

$$K = \frac{\lambda}{r_p}, \tag{14}$$

where $r_p$ refers to the particle radius. The Cunningham slip-flow correction is calculated from

$$G = 1 + K \left[ 1.249 + 0.42 \exp\left( -\frac{0.87}{K} \right) \right]. \tag{15}$$

Finally, the sedimentation velocity is obtained by means of Stokes law and from the slip-flow correction,

$$v_s = \frac{2r_p^2 g\,(\rho_p - \rho)}{9\eta} G, \tag{16}$$

with particle density $\rho_p$ and conventional standard gravitational acceleration $g$. Note that $r_p$ and $\rho_p$ can be specified individually for each air parcel. A larger set of parcels can be used to represent a size distribution of aerosol or cloud particles.

Figure 5 shows sedimentation velocities calculated by Eq. (16) for different particle diameters as well as pressure levels and mean temperatures at middle latitude atmospheric conditions. A standard particle density of $\rho_p = 1000\ \mathrm{kg\,m^{-3}}$ was considered. The sedimentation velocity $v_s$ scales nearly linearly with $\rho_p$, i. e., increasing $\rho_p$ by a factor of two will also increase $v_s$ by the same factor. Note that the overall sensitivity on temperature $T$ is much weaker than the sensitivity on pressure $p$ (not shown). Sedimentation velocities are shown here up to particle Reynolds number $Re_p \leq 1$, for which $v_s$ is expected to have accuracies better than 10 % (Hesketh, 1996). Larger particles will require additional corrections, which have not been implemented.

### 2.3.6 Wet deposition

Wet deposition causes the removal of trace gases and aerosol particles from the atmosphere within or below clouds by mixing with suspended water and following washout through rain, snow, or fog. In the first step, it is determined whether an air parcel is located below a cloud top. The cloud top pressure $p_{ct}$ is determined from the meteorological data as the highest vertical level where cloud water or ice (i. e., CLWC, CRWC, CIWC, or CSWC) is existent. For numerical efficiency, the level $p_{ct}$ is inferred at the time when the meteorological input data are processed. During the model run, $p_{ct}$ can therefore be determined quickly by interpolation from the preprocessed input data. Likewise, the cloud bottom pressure $p_{cb}$ is determined quickly from precalculated data.

In the second step, the wet deposition parametrization determines an estimate of the subgrid-scale precipitation rate $I_s$, which is needed to calculate the scavenging coefficient $\Lambda$. We estimated $I_s$ (in units of $\mathrm{mm\,h^{-1}}$) from the total column cloud water $c_l$ (in units of $\mathrm{kg\,m^{-2}}$) by means of a correlation function reported by Pisso et al. (2019),

$$I_s = (2\,c_l)^{1/0.36}, \tag{17}$$



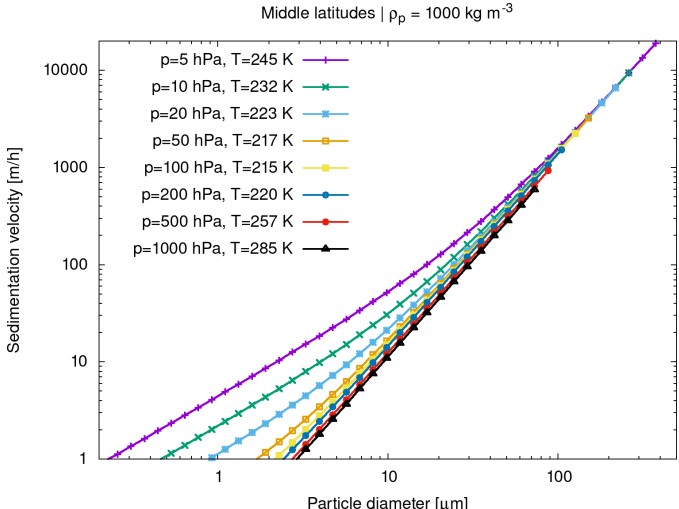

**Figure 5.** Sedimentation velocities of spherical particles in the Stokes-Cunningham regime for different particle sizes, atmospheric pressure levels, and middle latitudes mean temperatures.

which is based on a regression analysis using existing cloud and precipitation data. At the same time as determining $p_{ct}$ and $p_{cb}$, the total column cloud water $c_l$ is calculated by vertically integrating the cloud water content data when the meteorological input data are processed. During the model run, $c_l$ can therefore also be determined quickly by horizontal interpolation of the preprocessed input data to the locations of the air parcels. For efficiency, the parametrization of wet deposition is no further evaluated for a lower limit of $I_s < 0.01\,\mathrm{mm\,h^{-1}}$.

In the third step, it is inferred whether the air parcel is located within or below the cloud because scavenging coefficients will be different under these conditions. The position of the air parcel within or below the cloud is determined by interpolating the cloud water content to the position of the air parcel and by testing whether the interpolated values are larger than zero.

In the fourth step, the scavenging coefficient $\Lambda$ is calculated based on the precipitation rate $I_s$,

$$\Lambda = a\,I_s^b. \tag{18}$$

For aerosol particles, the constants $a$ and $b$ need to be specified as control parameters for the "within cloud" and "below cloud" cases, respectively. Typical values of $a = 4\ldots 8 \times 10^{-5}$ and $b = 0.79$ are used in the HYSPLIT and NAME models (Draxler and Hess, 1997; Webster and Thomson, 2014). For gases, wet deposition depends upon their solubility and the scavenging coefficient can be calculated from

$$\Lambda = H\,R\,T\,I_s\,\Delta z_c^{-1}, \tag{19}$$

where $H$ is Henry's law constant, $R$ is the universal gas constant, and $\Delta z_c$ is the depth of the cloud layer, which we calculate from $p_{ct}$ and $p_{cb}$. Henry's law constant is obtained from

$$H(T) = H^{\ominus} \exp\left[-\frac{\Delta_{sol}H}{R}\left(\frac{1}{T} - \frac{1}{T^{\ominus}}\right)\right]. \tag{20}$$





**Table 1.** Henry's law constants for water as a solvent. See Sander (2015) for reference.

| Formula | $H^{\ominus}$ (at 298.15 K) [mol m$^{-3}$ Pa$^{-1}$] | $-\frac{\Delta_{sol}H}{R}$ [K] |
|---|---|---|
| CF$_2$Cl$_2$ | $3.0 \times 10^{-5}$ | 3500 |
| CFCl$_3$ | $1.1 \times 10^{-4}$ | 3300 |
| CH$_4$ | $1.4 \times 10^{-5}$ | 1600 |
| CO | $9.7 \times 10^{-6}$ | 1300 |
| CO$_2$ | $3.3 \times 10^{-4}$ | 2400 |
| N$_2$O | $2.4 \times 10^{-4}$ | 2600 |
| NH$_3$ | $5.9 \times 10^{-1}$ | 4200 |
| HNO$_3$ | $2.1 \times 10^{3}$ | 8700 |
| NO | $1.9 \times 10^{-5}$ | 1600 |
| NO$_2$ | $1.2 \times 10^{-4}$ | 2400 |
| O$_3$ | $1.0 \times 10^{-4}$ | 2800 |
| SF$_6$ | $2.4 \times 10^{-6}$ | 3100 |
| SO$_2$ | $1.3 \times 10^{-2}$ | 2900 |

The constants $H^{\ominus}$ and $\frac{\Delta_{sol}H}{R}$ with enthalpy of dissolution $\Delta_{sol}H$ at the reference temperature $T^{\ominus} = 298.15$ K need to be specified as control parameters. Values for a wide range of species are tabulated by Sander (2015). The values of selected species of interest are summarized in Table 1 and included as default parameters in MPTRAC.

Finally, once the scavenging coefficient $\Lambda$ for the gases or aerosol particles within or below a cloud is determined, an exponential loss of mass $m$ over the time step $\Delta t$ is calculated,

$$m(t + \Delta t) = m(t) \exp(-\Delta t \Lambda). \tag{21}$$

### 2.3.7 Dry deposition

Dry deposition leads to a loss of mass of aerosol particles or trace gases by gravitational settling or chemical and physical interactions with the surface in the dry phase. In the parametrization implemented in MPTRAC, dry deposition is calculated for air parcels located in the lowermost $\Delta p = 30$ hPa layer above the surface. This corresponds to a layer width of $\Delta z \approx 200$ m

at standard conditions.

For aerosol particles, the deposition velocity $v_{dep}$ will be calculated as described in Sect. 2.3.5 as a function of surface pressure $p$ and temperature $T$ as well as particle radius $r_p$ and particle density $\rho_p$. For trace gases, the deposition velocity $v_{dep}$ needs to be specified as a control parameter. Currently, this parameter is set to a constant value across the globe for each trace gas. For future applications with a stronger focus on the boundary layer $v_{dep}$ will need to vary geographically to account for

dependence on the surface characteristics.





**Table 2.** Bimolecular reaction rate coefficients for the hydroxyl radical. See Burkholder et al. (2019) for reference.

| Reaction | $A$-factor | $E/R$ |
|----------|-----------|-------|
| $CH_4 + OH \rightarrow CH_3 + H_2O$ | $2.45 \times 10^{-12}$ | 1775 |
| $NH_3 + OH \rightarrow H_2O + NH_2$ | $1.7 \times 10^{-12}$ | 710 |
| $O_3 + OH \rightarrow HO_2 + O_2$ | $1.7 \times 10^{-12}$ | 940 |

For both, particles and gases, we calculated the loss of mass based on the deposition velocity $v_{dep}$, the model time step $\Delta t$, and the surface layer width $\Delta z$ from

$$m(t + \Delta t) = m(t) \exp\left(-\frac{\Delta t \, v_{dep}}{\Delta z}\right). \tag{22}$$

Note that both, the dry deposition and the wet deposition parametrization were implemented only very recently and need
further testing and evaluation in future work. They are described here for completeness, but they were not included in the model verification and performance analysis in Sect. 3.

### 2.3.8 Hydroxyl chemistry

In this section, we discuss the MPTRAC module that is used to simulate the loss of mass of a chemical species by means of reaction with the hydroxyl radical. The hydroxyl radical (OH) is an important oxidant in the atmosphere, causing the
decomposition of many gas phase species. The oxidation of different gas phase species with OH can be classified into two main categories, bimolecular reactions (e. g., reactions of $CH_4$ or $NH_3$) and termolecular reactions (e. g., CO or $SO_2$).

For bimolecular reactions, the rate constant is calculated from Arrhenius law,

$$k(T) = A \times \exp\left(-\frac{E}{RT}\right) \tag{23}$$

where $A$ refers to the Arrhenius factor, $E$ to the activation energy, and $R$ to the universal gas constant. For bimolecular
reactions, the Arrhenius factor $A$ and the ratio $E/R$ need to be specified as control parameters. The reaction rate coefficients for various gas phase species can be found in the Jet Propulsion Laboratory (JPL) data evaluation (Burkholder et al., 2019). Table 2 lists bimolecular reaction rate coefficients for some species of interest, which were also implemented as default values directly into MPTRAC.

Termolecular reactions require an inert component M (e. g., $N_2$ or $O_2$) to stabilize the excited intermediate state. Therefore,
the reaction rate shows a temperature and pressure dependence and changes smoothly between the low-pressure-limiting rate $k_0$ and high-pressure-limiting rate $k_\infty$. Based on the molecular density of air,

$$[M] = \frac{N_A \, p}{RT}. \tag{24}$$

with Avogadro constant $N_A$, the effective second-order order rate constant $k$ is calculated from

$$k(T, [M]) = \frac{k_0(T)\,[M]}{1 + \frac{k_0(T)\,[M]}{k_\infty(T)}} \, 0.6^{\left\{1 + \left[\log_{10}\left(\frac{k_0(T)\,[M]}{k_\infty(T)}\right)\right]^2\right\}^{-1}}. \tag{25}$$





**Table 3.** Termolecular reaction rate coefficients for the hydroxyl radical. See Burkholder et al. (2019) for reference.

| Reaction | $k_0^{298}$ | $n$ | $k_\infty^{298}$ | $m$ |
|---|---|---|---|---|
| $CO + OH \xrightarrow{M} HOCO$ | $6.9 \times 10^{-33}$ | 2.1 | $1.1 \times 10^{-12}$ | $-1.3$ |
| $NO + OH \xrightarrow{M} HONO$ | $7.1 \times 10^{-31}$ | 2.6 | $3.6 \times 10^{-11}$ | $0.1$ |
| $NO_2 + OH \xrightarrow{M} HONO_2$ | $1.8 \times 10^{-30}$ | 3.0 | $2.8 \times 10^{-11}$ | $0$ |
| $SO_2 + OH \xrightarrow{M} HOSO_2$ | $2.9 \times 10^{-31}$ | 4.1 | $1.7 \times 10^{-12}$ | $-0.2$ |

The low- and high-pressure limits of the reaction rate constant are given by

$$k_0(T) = k_0^{298} \left( \frac{T}{298} \right)^{-n},$$ (26)

$$k_\infty(T) = k_\infty^{298} \left( \frac{T}{298} \right)^{-m}.$$ (27)

The constants $k_0^{298}$ and $k_\infty^{298}$ at the reference temperature of 298 K and the exponents $n$ and $m$ need to be specified as control parameters. The exponents can be set to zero in order to neglect the temperature dependence of the low- or high-pressure limits

of $k_0$ and $k_\infty$. The termolecular reaction rate coefficients implemented directly into MPTRAC are listed in Table 3.

Based on the bimolecular reaction rate $k = k(T)$ from Eq. (23) or the termolecular reaction rate $k = k(T, [M])$ from Eq. (25), the loss of mass of the gas phase species over time is calculated from

$$m(t + \Delta t) = m(t) \exp(-k [OH] \Delta t).$$ (28)

The hydroxyl radical concentrations [OH] are obtained by bilinear interpolation from the monthly mean zonal mean clima-

tology of Pommrich et al. (2014). This approach is suitable for global simulations covering at least several days, as hydroxyl concentrations may vary significantly between day- and nighttime as well as the local atmospheric composition.

### 2.3.9 Exponential decay

A rather generic module was implemented in MPTRAC, to simulate the loss of mass of an air parcel over a model time step $\Delta t$ due to any kind of exponential decay process, e. g., chemical loss or radioactivity,

$$m(t + \Delta t) = m(t) \exp \left( -\frac{\Delta t}{t_e} \right).$$ (29)

The e-folding lifetime $t_e$ of the species needs to be specified as a control parameter. As typical lifetimes may differ, we implemented an option to specify separate lifetimes for the troposphere and stratosphere. A smooth transition between the tropospheric and stratospheric lifetime is created within a $\pm 1$ km log-pressure altitude range around the tropopause.

### 2.3.10 Boundary conditions

Finally, a module was implemented to impose constant boundary conditions on particle mass or volume mixing ratio. At each time step of the model, all particles that are located within a given latitude and pressure range can be assigned a constant mass





or volume mixing ratio as defined by a control parameter of the model. Next to specifying fixed pressure ranges in the free troposphere, stratosphere, or mesosphere to define the boundary conditions, it is also possible to specify a fixed pressure range with respect to the surface pressure, in order to define a near-surface layer.

The boundary condition module can be used to conduct synthetic tracer simulations. For example, the synthetic tracer E90 (Prather et al., 2011; Abalos et al., 2017) is emitted uniformly at the surface and has a constant e-folding lifetime of 90 days throughout the entire atmosphere. In our study, we implemented this by assigning a constant volume mixing ratio of 150 ppb to all air parcels residing in the lowermost 30 hPa pressure range above the surface. With its 90-day lifetime, the tracer E90 quickly becomes well-mixed in the troposphere. However, the lifetime is much shorter than typical timescales

of stratospheric transport. The tracer E90 therefore exhibits sharp gradients across the tropopause. Being a passive tracer in the upper troposphere and lower stratosphere region, E90 is evaluated in studies related to the chemical tropopause and stratosphere-troposphere exchange.

Another example of a synthetic tracer is ST80, which is defined by a constant volume mixing ratio of 200 ppb above 80 hPa and a uniform, fixed 25 day e-folding lifetime in the troposphere (Eyring et al., 2013). Like E90, the tracer ST80 is of interest

in investigating stratosphere-troposphere exchange. The Northern hemisphere synthetic tracer NH50 is considered to assess interhemispheric gradients in the troposphere. It is defined by a surface layer volume mixing ratio of 100 ppb over 30 to 50°N and a uniform 50 day e-folding lifetime (Eyring et al., 2013). Simulation results of E90, ST80, and NH50 obtained with the CPU and GPU code of MPTRAC will be discussed in Sect. 3.3.

## 2.4   Model output data

### 480   2.4.1   Sampling of meteorological data

For diagnostic purposes, it is often necessary to obtain meteorological data along the trajectories. However, typically only a limited subset of the meteorological variables is of interest. Therefore, we introduced the concept of "quantities", allowing the user to select, which variables should be calculated and stored in the output data files. Next to selecting the specific types of model output that are desired, also a list of the specific quantities that should be provided as output needs to be specified via

control parameters. In total, about 50 output variables are implemented in MPTRAC (Table 4). In addition to the predefined output variables, it is possible to include user-defined variables. The user-defined variables are typically not modified during the simulations, but they are useful to store information that are required for further analyses. For example, the starting time and position of the particles can be stored as user-defined variables, and they can later be used to calculate the trajectory time or the distance from the origin.

A distinct module was implemented into MPTRAC that is used to sample the meteorological data along the trajectories. This module can become demanding in terms of computing time, as it requires the interpolation of up to ten 3-D variables and sixteen 2-D variables of meteorological data that are either read in from the meteorological input data files or calculated during the preprocessing of the meteorological data. However, the computing time needs of this module are typically strongly





**Table 4.** List of quantities defined in MPTRAC.

| Identifier | Symbol | Description | Units |
|---|---|---|---|
| ENS | – | ensemble index | – |
| STAT | – | station visit flag | – |
| M | $m$ | mass of species | kg |
| VMR | $x$ | volume mixing ratio | ppv |
| R | $r_p$ | particle radius | $\mu$m |
| RHO | $\rho_p$ | particle density | $\mathrm{kg\,m^{-3}}$ |
| PS | $p_s$ | surface pressure | hPa |
| TS | $T_s$ | surface temperature | K |
| ZS | $Z_s$ | surface geopotential height | km |
| US | $u_s$ | surface zonal wind | m/s |
| VS | $v_s$ | surface meridional wind | m/s |
| PBL | $p_b$ | planetary boundary layer pressure | hPa |
| Z | $Z$ | geopotential height | km |
| P | $p$ | pressure | hPa |
| T | $T$ | temperature | K |
| U | $u$ | zonal wind | $\mathrm{m\,s^{-1}}$ |
| V | $v$ | meridional wind | $\mathrm{m\,s^{-1}}$ |
| W | $\omega$ | vertical velocity | $\mathrm{hPa\,s^{-1}}$ |
| VH | $v_h$ | horizontal wind velocity | $\mathrm{m\,s^{-1}}$ |
| VZ | $v_z$ | vertical velocity | $\mathrm{m\,s^{-1}}$ |
| PV | $Q$ | potential vorticity | PVU |
| H2O | $x_{\mathrm{H_2O}}$ | water vapor volume mixing ratio | ppv |
| HNO3 | $x_{\mathrm{HNO_3}}$ | nitric acid volume mixing ratio | ppv |
| O3 | $x_{\mathrm{O_3}}$ | ozone volume mixing ratio | ppv |
| OH | $n_{\mathrm{OH}}$ | hydroxyl number density | $\mathrm{molec\,cm^{-3}}$ |
| THETA | $\theta$ | potential temperature | K |
| ZETA | $\zeta$ | hybrid vertical coordinate | K |
| TVIRT | $T_v$ | virtual temperature | K |
| LAPSE | $\Gamma$ | moist adiabatic lapse rate | $\mathrm{K\,km^{-1}}$ |

reduced by the fact that meteorological data along the trajectories are usually not required at every model time step but only at

user-defined output intervals.

Next to the option of sampling of the meteorological data along the trajectories, we also implemented tools to provide direct output of the meteorological data. This includes tools to extract maps, zonal mean cross-sections, and vertical profiles





| Identifier | Symbol | Description | Units |
|---|---|---|---|
| PT | $p_t$ | tropopause pressure | hPa |
| TT | $T_t$ | tropopause temperature | K |
| ZT | $Z_t$ | tropopause geopotential height | km |
| H2OT | $x_{\mathrm{H_2O},t}$ | tropopause water vapor volume mixing ratio | ppv |
| PSAT | $p_{sat}$ | saturation pressure over water | hPa |
| PSICE | $p_{\mathrm{ice}}$ | saturation pressure over ice | hPa |
| PW | $p_w$ | partial water vapor pressure | hPa |
| SH | $q$ | specific humidity | $\mathrm{kg\,kg^{-1}}$ |
| RH | RH | relative humidity over water | % |
| RHICE | $\mathrm{RH_{ice}}$ | relative humidity over ice | % |
| LWC | CLWC | cloud liquid water content | $\mathrm{kg\,kg^{-1}}$ |
| IWC | CIWC | cloud ice water content | $\mathrm{kg\,kg^{-1}}$ |
| CL | $c_l$ | total column cloud water | $\mathrm{kg\,m^{-2}}$ |
| PCT | $p_{ct}$ | cloud top pressure | hPa |
| PCB | $p_{cb}$ | cloud bottom pressure | hPa |
| PLCL | $p_{\mathrm{LCL}}$ | pressure at lifted condensation level | hPa |
| PLFC | $p_{\mathrm{LFC}}$ | pressure at level of free convection | hPa |
| PEL | $p_{\mathrm{EL}}$ | pressure at equilibrium level | hPa |
| CAPE | CAPE | convective available potential energy | $\mathrm{J\,kg^{-1}}$ |
| TDEW | $T_{\mathrm{dew}}$ | dew point temperature | K |
| TICE | $T_{\mathrm{ice}}$ | frost point temperature | K |
| TNAT | $T_{\mathrm{NAT}}$ | NAT particle existence temperature | K |
| TSTS | $T_{\mathrm{STS}}$ | STS particle existence temperature | K |

on pressure or isentropic levels. These tools allow for time averaging over multiple meteorological data files. Another tool is available to sample the meteorological data based on a list of given times and positions, without involving any trajectory 500 calculations. Altogether, these tools provide a great flexibility in interpolating meteorological data for many applications.

### 2.4.2 Output from Lagrangian transport simulations

At present, MPTRAC offers seven output options, referred to as "atmospheric output", "grid output", "CSI output", "ensemble output", "profile output", "sample output", and "station output". The most comprehensive output of MPTRAC is the atmospheric output. Atmospheric output files can be generated at user-defined time intervals, which need to be integer multiples of





the model time step $\Delta t$. The atmospheric output files are the most comprehensive type of output because they contain the time, location, and the values of all user-defined quantities of each individual air parcel.

As the atmospheric output can easily become too large for further analyses, in particular if many air parcels are involved, the output of gridded data was implemented. This output will be generated by integrating over the mass of all parcels in regular longitude × latitude × log-pressure height grid boxes. From the total mass per grid box and the air density, the column density

and the volume mixing ratio of the tracer are calculated. Alternatively, if the volume mixing ratio per air parcel is specified, the mean volume mixing ratio per grid box is reported. In the vertical, it is possible to select only a single layer for the grid output, in order to obtain total column data. Similarly, by selecting only one grid box in longitude, it is possible to calculate zonal means.

Another type of output that we used in several studies (Hoffmann et al., 2016; Heng et al., 2016) is the critical success

index (CSI) output. This output is produced by analyzing model and observational data on a regular grid. The analysis is based on a $2 \times 2$ contingency table of model predictions and observations. Here, predictions and observations are counted as *yes*, if the model column density or the observed variable exceed user-defined thresholds. Otherwise, they would be counted as *no*. Next to the CSI, the counts allow us to calculate the probability of detection (POD) and the false alarm rate (FAR), which are additional skill scores that are often considered in model verification. More recently, the CSI output was extended to

also include the equitable threat score (ETS), the linear and rank-order correlation coefficients, the bias, the root mean square (RMS) difference, and the mean absolute error. A more detailed discussion of the skill scores is provided by Wilks (2011).

Another option to condense comprehensive particle data is provided by means of the ensemble output. This type of output requires a user-defined specific ensemble index value to be assigned to each air parcel. Instead of the individual air parcel data, the ensemble output will contain the mean positions as well as the means and standard deviations of the quantities selected for

output for each set of air parcels having the same ensemble index. The ensemble output if of interest, if tracer dispersion from multiple point sources needs to be quantified by means of a single model run, for instance.

The profile output of MPTRAC is similar to the grid output as it creates vertical profiles from the model data on a regular longitude × latitude horizontal grid. However, the vertical profiles do not only contain volume mixing ratios of the species of interest, but also profiles of pressure, temperature, water vapor, and ozone as inferred from the meteorological input data. This

output is compiled with the intention to be used as input for a radiative transfer model, in order to simulate satellite observations for the given model output. In combination with real satellite observations, this output can be used for model validation but also as a basis for radiance data assimilation.

The sample output of MPTRAC was implemented most recently. It allows the user to extract model information on a list of given locations and times, by calculating the column density and volume mixing ratio of all parcels located within a user-

specified horizontal search radius and vertical height range. For large numbers of sampling locations and air parcels, this type of output can become rather time-consuming. It requires an efficient implementation and parallelization because it needs to be tested at each time step of the model whether an air parcel is located within a sampling volume or not. The numerical effort scales linearly with both, the number of air parcels and the number of sampling volumes. The sample output was first applied in




the study of Cai et al. (2021) to sample MPTRAC data directly on TROPOspheric Monitoring Instrument (TROPOMI) satellite
observations.

Finally, the station output is collecting the data of air parcels that are located within a search radius around a given location
(latitude, longitude). The vertical position is not considered here, i. e., the information of all air parcels within the vertical
column over the station is collected. In order to avoid double-counting of air parcels over multiple time steps, the quantity STAT
(Table 4) has been introduced that keeps track on whether an air parcel has already been accounted for in the station output
or not. We used this type of output in studies estimating volcanic emissions from satellite observations using the backward
trajectory method (Hoffmann et al., 2016; Wu et al., 2017, 2018).

By default, all output functions of MPTRAC create data files in an ASCII table format. This type of output is usually simple
to understand and usable with many tools for data analysis and visualization. However, in case of large-scale simulations it is
desirable to use more efficient file formats. Therefore, an option was implemented to write particle data to binary output files.
Likewise, reading particle data from a binary file is much more efficient than from an ASCII file. The binary input and output is
an efficient way to save or restore the state of the model during intermediate steps of a workflow. Another interesting option of
output is to pipe the data directly from the model to a visualization tool. This will keep the output data in memory and directly
forward it from MPTRAC to the visualization tool. This option has been successfully tested for the particle and the grid output
in combination with the graphing utility `gnuplot` (Williams and Kelley, 2020).

### 2.4.3 Trajectory statistics

For diagnostic purposes, it is often helpful to calculate trajectory statistics. The tool `atm_stat` can calculate the mean,
standard deviation, skewness, kurtosis, median, absolute deviation, median absolute deviation, minimum, and maximum, of
the positions and variables of air parcel data sets. The calculations can cover the entire set or a selection of air parcels, based
on the ensemble variable or extracted by means of the tool `atm_select`. The tool `atm_select` allows to select air parcels
based on a range of parcels, a log-pressure height × longitude × latitude box, and a search radius around a given geolocation.
The trajectory statistics are calculated by applying the algorithms of the GNU Scientific Library (Gough, 2003).

The tool `atm_dist` allows us to calculate statistics of the deviations between two sets of trajectories. Foremost, this
includes the absolute horizontal and vertical transport deviations (AHTD and AVTD, Kuo et al., 1985; Rolph and Draxler,
1990; Stohl, 1998) over time,

$$AHTD(t_n) = \frac{1}{N} \sum_{i=1}^{N} \sqrt{[x_i(t_n) - X_i(t_n)]^2 + [y_i(t_n) - Y_i(t_n)]^2}, \tag{30}$$

$$AVTD(t_n) = \frac{1}{N} \sum_{i=1}^{N} |z_i(t_n) - Z_i(t_n)|. \tag{31}$$

Here, the horizontal distances are calculated approximately by converting the geographic longitudes and latitudes of the particles to Cartesian coordinates, $[x_i(t_n), y_i(t_n), z_i(t_n)]$ and $[X_i(t_n), Y_i(t_n), Z_i(t_n)]$, followed by calculation of the Euclidean
distance of the Cartesian coordinates. Vertical distances are calculated based on conversion of particle pressure to log-pressure
altitude using the barometric formula.



Next to AHTDs and AVTDs, the tool `atm_dist` can calculate the relative horizontal and vertical transport deviations (RHTD and RVTD), which relate the absolute transport deviations to the horizontal and vertical path lengths of individual trajectories. It can also be used to quantify the mean absolute and relative deviations as well as the absolute and relative bias of meteorological variables or the relative tracer conservation errors of dynamical tracers over time. The definitions of these quantities and more detailed descriptions are provided by Hoffmann et al. (2019). In this study, we mostly present absolute transport deviations, to evaluate the differences between CPU and GPU trajectory calculations (Sect. 3.2).

## 2.5 Parallelization strategy

Lagrangian particle dispersion models are well suited for parallel computing as large sets of air parcel trajectories can be calculated independently of each other. The workload is "embarrassingly parallel" or "perfectly parallel" as little to no effort is needed to separate the problem into a number of independent parallel computing tasks. In this section, we discuss the MPI/OpenMP/OpenACC hybrid parallelization implemented in MPTRAC. The term "hybrid parallelization" refers to the fact that several parallelization techniques are employed in a simulation at the same time.

The Message Passing Interface (MPI) is a communication protocol and standardized interface to enable point-to-point and collective communication between computing tasks. MPI provides high performance, scalability, and portability. It is considered as a leading approach used for high-performance computing. The application programming interface Open Multi-Processing (OpenMP) supports multi-platform shared-memory multiprocessing programming for various programming languages and computing architectures. It consists of a set of compiler directives, library routines, and environment variables that are used to distribute the workload over a set of computing threads on a compute node. An application built with an MPI/OpenMP hybrid approach can run on a compute cluster, such that OpenMP can be used to exploit the parallelism of all hardware threads within a multi-core node while MPI is used for parallelism between the nodes. Open accelerators (OpenACC) is a programming model to enable parallel programming of heterogeneous CPU/GPU systems. As in OpenMP, the source code of the program is annotated with "pragmas" to identify the areas that should be accelerated by using GPUs. In combination with MPI and OpenMP, OpenACC can be used to conduct multi-GPU simulations.

The GPU parallelization is a new feature since version 2.0 of MPTRAC. Various concepts for employing GPUs are available for scientific high-performance computing. The simplest option is to replace standard computing libraries by GPU-enabled versions, which are provided by the vendors of the GPU hardware such as the NVIDIA company. A prominent example of such a library, which was ported to GPUs, is the Basic Linear Algebra Subprograms (BLAS) library for matrix-matrix, matrix-vector, and vector-vector operations. However, as MPTRAC does not employ the BLAS library, this simple parallelization technique was not considered helpful for porting our model to GPUs. At the other end of the options for GPU computing, the Compute Unified Device Architecture (CUDA) is a dedicated application programming model for GPUs. The CUDA programming model is often considered to be most flexible and allows for most detailed control over the GPU hardware. However, CUDA requires solid knowledge on GPU technology and it may cause error-prone and time-consuming work if legacy code needs to be rewritten (Li and Shih, 2018).





OpenACC can help to overcome the practical difficulties of CUDA and reduces the coding workload for developers signifi-
cantly. We considered OpenACC to be a more practical choice for porting MPTRAC to GPUs as the code of the model should
remain understandable and maintainable by students and domain scientists that may not be familiar with the details of more
complex GPU programming concepts such as CUDA. By using OpenACC, we found that we were able to maintain the same
code base of the MPTRAC model rather than having to develop a CPU and a GPU version of the model independently of each
other. Next to easiness in terms of maintenance and portability of the code, the common code base is considered a significant
advantage to check for bit-level agreement of CPU and GPU computations and reproducibility of simulation results.

Algorithm 1 illustrates the GPU porting of the C code of MPTRAC by means of OpenACC pragmas. Two important aspects
need to be considered. First, the pragma `#pragma acc parallel loop independent gang vector` is used to
distribute the calculations within the loops over the particles over the compute elements of the GPU device. In this example,
the loop over the particles within the advection module used to calculate the trajectories is offloaded to GPUs. For compute
kernels that are used inside of a GPU parallelized code block, the compiler will create both, a CPU and a GPU version of
the kernel. In this example, the pragma `#pragma acc routine (intpol_met_time_3d)` instructs the compiler to
create CPU and GPU code for the function used to interpolate meteorological data, as used within and outside of the advection
module. Algorithm 1 also shows that if the code is compiled without GPU support (as identified by the _OPENACC flag), the
fall-back option is to apply OpenMP parallelization for the compute loops on the CPUs (`#pragma omp parallel for
default(shared)`). The OpenMP parallelization was already implemented in an earlier, CPU-only version of MPTRAC.

The second aspect of GPU porting by means of OpenACC is concerned with the data management. In principle, the data
required for a calculation need to be copied from CPU to GPU memory before the calculation and need to be copied back from
GPU to CPU memory after the computation finished. Although NVIDIA's "CUDA Unified Memory" technique makes sure,
these data transfers can be done automatically, frequent data transfers can easily become a bottleneck of the code. Therefore,
we implemented additional pragmas to instruct the compiler when a data transfer is actually required.

The pragma `#pragma acc enter data create(ctl,atm[:1],met0[:1],met1[:1])` in Algorithm 1 cre-
ates a data region and allocates memory for the model control parameters (`ctl`), air parcel data (`atm`), and meteorological
data (`met0`, `met1`) on the GPU. The data region is deleted and the GPU memory is freed by `#pragma acc exit data
delete(ctl,atm,met0,met1)` at the end of the main function. Within the data region, the associated data will remain
present in GPU memory. Within the data region, the pragma `#pragma acc update device(atm[:1],ctl)` is used
to explicitly copy data from CPU to GPU memory, whereas `#pragma acc update host(atm[:1])` is used to copy
data from GPU to CPU memory. In the MPTRAC model, a single large data region was created around the main loop over the
time steps of the model. As a consequence, the calls to all physics and chemistry modules will be conducted completely on
GPU memory. Data transfers between CPUs and GPUs are needed only for updating of the meteorological data or for writing
model output.

In total, about 60 pragmas had to be implemented in the code to parallelize the computational loops and to handle the data
management to facilitate the GPU porting of MPTRAC by means of OpenACC. Code parts that were ported to GPUs are
highlighted in the call graph shown in Fig. 1. Next to the OpenACC pragmas, some additional code changes were necessary



---

**Algorithm 1** GPU porting of MPTRAC by means of OpenACC pragmas.

---

```
/* Main function of the MPTRAC model... */
int main(int argc, char *argv[]) {
  ...
  /* Initialize model run... */
  ...
  /* Create data region on GPUs... */
  #pragma acc enter data create(ctl,atm[:1],met0[:1],met1[:1])
  ...
  /* Read initial air parcel data... */
  read_atm(filename, &ctl, atm);
  #pragma acc update device(atm[:1],ctl)
  ...
  /* Loop over timesteps... */
  for (t = t_start; t < t_stop; t += dt) {
    ...
    /* Read meteorological data... */
    get_met(&ctl, t, &met0, &met1);
    #pragma acc update device(met0[:1],met1[:1],ctl)
    ...
    /* Calculate advection... */
    module_advection(met0, met1, atm, dt);
    ...
    /* Write output... */
    #pragma acc update host(atm[:1])
    write_output(dirname, &ctl, met0, met1, atm, t);
  }
  ...
  /* Delete data region on GPUs... */
  #pragma acc exit data delete(ctl,atm,met0,met1)
  ...
}
```

---

to enable the GPU porting. In the original CPU code, the GNU Scientific Library (GSL) was applied to conduct a number of

tasks, for instance, to compute statistics of data arrays. As the GSL is not ported to GPUs, the corresponding functions had to

be rewritten without usage of the GSL. The GSL was also used to generate random numbers. Here, we implemented NVIDIA's



---

**Algorithm 1** Continued.

---

```
/* Function used to calculate air parcel trajectories... */
void module_advection(met_t * met0, met_t * met1, atm_t * atm, double *dt) {

  /* Loop over air parcels... */
  #ifdef _OPENACC
  #pragma acc data present(met0,met1,atm,dt)
  #pragma acc parallel loop independent gang vector
  #else
  #pragma omp parallel for default(shared)
  #endif
  for (int ip = 0; ip < atm->np; ip++) {
    ...
    /* Interpolate meteorological data... */
    intpol_met_time_3d(met0, met1, p, lon, lat...);
    ...
    /* Get position of the mid point... */
    ...
    }
}

/* Function used to interpolate meteo data... */
#pragma acc routine (intpol_met_time_3d)
void intpol_met_time_3d(met0, met1, p, lon, lat...) {
  ...
}
```

---

CUDA random number generation library (cuRAND) as an efficient replacement for distributed random number generation on GPUs. Also, we found that some standard math library functions were not available with PGI's C compiler, e. g., a replacement had to be implemented for the math function fmod used to calculate floating point modulo. For earlier versions of the PGI
C compiler, we also found a severe bug when large data structures ($>2\,\mathrm{GByte}$) were transferred from CPU to GPU memory. Nevertheless, despite a number of technical issues, we consider the porting of MPTRAC to GPUs by means of OpenACC to be straight forward and would recommend it for other applications.



## 3 Model verification and performance analysis

### 3.1 Description of the GPU test system and software environment

The model verification and performance analysis described in this section was conducted on the Jülich Wizard for European Leadership Science (JUWELS) system (Jülich Supercomputing Centre, 2019). The JUWELS system is considered as a major target system for running Lagrangian transport simulations with MPTRAC in future work. The JUWELS system is composed of a CPU module, referred to as the JUWELS Cluster, and a GPU module, referred to as the JUWELS Booster. The JUWELS Booster and the JUWELS Cluster can be used jointly as a heterogeneous, modular supercomputer, or they can be used inde-

pendently of each other. The JUWELS Booster was used as a test system in this study.

The JUWELS Booster consists of 936 compute nodes, each equipped with four NVIDIA A100 Tensor Core GPUs. Each NVIDIA A100 GPU comprises 6912 INT32 and 6912 FP32 compute cores as well as 3456 FP64 compute cores. It also features 432 tensor cores, providing $8\times$ peak speedup for mixed-precision matrix multiplication compared to standard FP32 precision operations. The numbers of compute elements listed here indicate the large degree of parallelism required for the

compute problem, which is exploited by large numbers of particles in the Lagrangian transport simulations. The A100 GPUs are equipped with 40 GByte HBM2e memory per device, connected with the third-generation NVLink lossless, high-bandwidth, low-latency shared memory interconnect, providing 1555 GByte s$^{-1}$ of memory bandwidth. The A100 GPU supports the new Compute Capability 8.0. NVIDIA (2020) provides a detailed description of the Ampere architecture.

The JUWELS Booster GPUs are hosted by AMD EPYC Rome 7402 CPUs. Each compute node of the Booster comprises two

CPU sockets, four non-uniform memory access (NUMA) domains per CPU, and six compute cores per NUMA domain, which provide two-way simultaneous multithreading. Up to 48 physical threads or 96 virtual threads can be executed on the CPUs of a compute node. The compute nodes are equipped with 512 GByte DDR4-3200 RAM. They are connected by a Mellanox HDR200 InfiniBand ConnectX 6 network in a DragonFly+ topology. CPUs, GPUs, and network adapters are connected via two PCIe Gen 4 switches with 16 PCIe lanes going to each device. A 350 GByte s$^{-1}$ network connection IBM Spectrum Scale

(GPFS) parallel file system is implemented. As of November 2020, the JUWELS Booster yields a peak performance (Rpeak) of 70.98 PFlop/s (see https://www.top500.org/lists/top500/2020/11/, last access: 21 May 2021).

The software environment on JUWELS is based on the CentOS 8 Linux distribution. Compute jobs are managed by the Slurm batch system with ParTec's ParaStation resource management. The JUWELS Booster software stack comprises the CUDA-aware ParTec ParaStation MPI and the NVIDIA HPC Software Development Kit (SDK), as used in this work. In

particular, we used the PGI C Compiler (PGCC) version 21.5, which more recently was rebranded and integrated as NVC in NVIDIA's HPC SDK, and the GNU C compiler (GCC) version 10.3.0 to compile the GPU and CPU code, respectively. We applied the compile flag for strong optimization (-O3) for both, GCC and PGCC. For other standard compile flags, please refer to the Makefile in the MPTRAC code repository. MPTRAC was build with the netCDF library version 4.7.4 and HDF5 library version 1.12.0 for file-I/O. The GNU Scientific Library (GSL) version 2.7 was used for generating random numbers on the

CPUs. The CUDA random number generation library (cuRAND) from the CUDA Toolkit release 11.0 was applied for random number generation on the GPU devices.





## 3.2 Comparison of kinematic trajectory calculations

In this section, we discuss comparisons of kinematic trajectory calculations conducted with the CPU and GPU versions of the MPTRAC model. For the CPU simulations, we considered binaries created with two different C compilers. The CPU code compiled with GCC is considered here as a reference, as this compiler has been used in most of the previous work and earlier studies with MPTRAC. The second version of the CPU binaries as well as the GPU binaries were compiled with PGCC, which is the recommended compiler for GPU applications on the JUWELS Booster.

We first focus on comparisons of kinematic trajectories that were calculated with horizontal wind and vertical velocity fields of the ERA5 reanalysis. The calculations were conducted without subgrid-scale wind fluctuations, turbulent diffusion, or convection, as these parametrizations rely on different random number generators implemented in the cuRAND and GSL libraries and utilize a different parallelization approach for random number generation for the CPU and GPU code, respectively. Individual trajectories calculated with these modules being activated are therefore not directly comparable to each other. They can only be compared in a statistical sense, as in Sect. 3.3.

We globally distributed the trajectory seeds in the pressure range from the surface up to 0.1 hPa (about $0-64$ km of log-pressure altitude). The starting time of the trajectories was 1 January 2017, 00:00 UTC, and the calculations cover 60 days of trajectory time. To evaluate the trajectories, we calculated absolute horizontal transport deviations (AHTDs), absolute vertical transport deviations (AVTDs), relative horizontal transport deviations (RHTDs), and relative vertical transport deviations (RVTDs) at 6-hourly time intervals (see Sect. 2.4.3). Considering that the trajectory errors often significantly depend on the altitude range of the atmosphere and to enable comparisons with earlier work (Rößler et al., 2018; Hoffmann et al., 2019), we calculated the transport deviations for log-pressure height ranges of $0-2$ km (boundary layer), $2-8$ km (lower to middle troposphere), $8-16$ km (middle troposphere to lower stratosphere), $16-32$ km (lower to middle stratosphere), $32-48$ km (middle to upper stratosphere), and $48-64$ km (mesosphere).

The absolute transport deviations between the GPU and CPU versions of MPTRAC with the GCC and PGCC compilers are shown in Fig. 6. Up to 30 days of trajectory time, the transport deviations between the different binaries and versions of the code are negligible, i. e., they are $\leq 1$ km for the AHTDs, and $\leq 1$ m for the AVTDs at all heights. This corresponds to RHTDs and RVTDs of $\leq 0.01\%$ (not shown). The small deviations between the different binaries during the first 30 days might be attributed to different compiler settings, which enforce aggressive optimizations to achieve performance improvements coming by the price of rounding errors in the math library functions. After about 30 days, the small rounding errors further accumulate so that after 60 days, the differences between the PGCC GPU and CPU code results increase to up to 120 km (0.3%) in the horizontal domain and about 70 m (0.4%) in the vertical domain. The smallest transport deviations are found in the lower stratosphere (16 to 32 km of altitude), which is consistent with earlier work showing particularly low deviations for this height range due to favorable background wind and velocity conditions (Hoffmann et al., 2019).

Note that the CPU and GPU binaries created by PGCC (colored curves in Fig. 6) show somewhat better agreement (by a factor of 3 to 5) compared to deviations between the CPU binaries created by GCC and PGCC (solid gray curves). These differences between the compilers may also be attributed to different optimization settings applied at compile time. However,

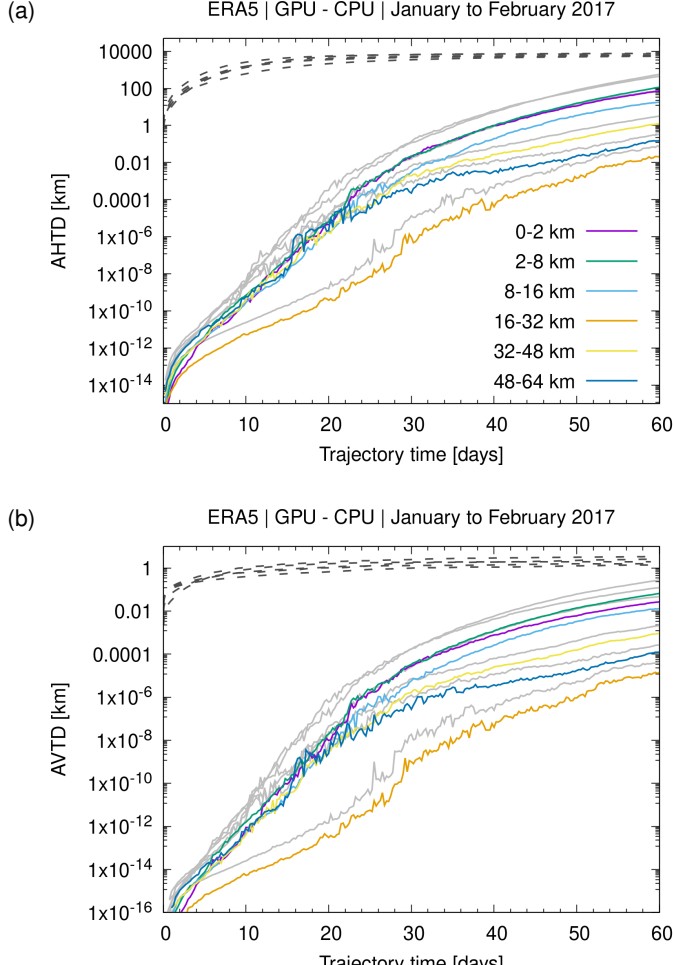

**Figure 6.** Absolute (a) horizontal and (b) vertical transport deviations (AHTDs and AVTDs) between kinematic trajectories calculated with the CPU and GPU code of MPTRAC as compiled with the PGI C compiler (colored curves). Simulations start on 1 January 2017, 00:00 UTC and comprise 60 days of simulation time. For this test case, $10^6$ trajectory seeds were globally distributed in different height ranges between the lower troposphere and the mesosphere (see plot key). Transport deviations between simulations using the CPU code compiled with the GNU and PGI C compilers are shown for comparison (light gray, solid curves). Transport deviations due to subgrid-scale wind fluctuations and diffusion are also shown for reference (dark gray, dashed curves).

overall, the transport deviations found here are several orders of magnitude smaller than deviations caused by other factors, such as deviations related to different numerical integration schemes and the choice of the model time step (Rößler et al., 2018) or the impact of subgrid-scale winds and diffusion (Hoffmann et al., 2019). Therefore, we consider this result to show excellent agreement between the kinematic trajectory calculations with the GPU and CPU code and the different compilers.





## 3.3 Comparison of synthetic tracer simulations

With our current code, we cannot directly compare individual trajectories including the effects of turbulent diffusion and subgrid-scale winds as well as convection because these modules add stochastic perturbations to the trajectories that are created by means of the different random number generators of the cuRAND library for the GPUs and the GSL library for the CPUs. In order to compare CPU and CPU simulations considering these modules, we conducted global transport simulations of synthetic tracers with ERA5 data. In these simulations, we included the tropospheric synthetic tracers E90 and NH50 as well as the stratospheric synthetic tracer ST80 (Sect. 2.3.10). The simulations were initialized with $10^6$ globally homogeneously distributed air parcels in the height range from the surface up to the lower mesosphere on 1 January 2017, 00:00 UTC. The simulations cover the entire time range of the year of 2017. We assigned volume mixing ratios to the individual air parcels according to the specific initial and boundary conditions of the synthetic tracers (Sect. 2.3.10). Expecting that differences between individual trajectories are largely cancelling out due to the averaging, we then compared monthly mean zonal means of the synthetic tracer volume mixing ratios from the CPU and GPU simulations.

As a representative example, Fig. 7c shows the monthly mean zonal mean distribution of E90 in July 2017 from our ERA5 simulations with the GPU code of MPTRAC. Following a spin-up time of half a year, i. e., about twice the lifetime of the E90 tracer, it is found that E90 is well-mixed and rather homogeneously distributed throughout the troposphere. The 90 ppbv contour of the E90 tracer is expected to indicate the chemical tropopause (Prather et al., 2011). For our simulations, we found that the 90 ppbv contour line of E90 agrees well with the monthly mean zonal dynamical tropopause. The E90 distribution in Fig. 7c qualitatively compares well with the climatological mean presented by Abalos et al. (2017).

Figure 7d shows the differences of the E90 monthly mean zonal means between the GPU and CPU code. Despite the fact that even without diffusion or convection being considered, individual CPU and GPU trajectories already differ significantly after 60 days (Sect. 3.2), the E90 monthly mean zonal means of the CPU and GPU code agree quite well. The differences of the means are typically below $\pm 8$ ppb, which is about 5 % of the maximum volume mixing ratio of the E90 tracer. The largest differences between the CPU and GPU simulations are found near the tropopause. This region is most sensitive to statistical effects, as the tracer gradients are largest in this region. However, no systematic bias between the E90 distributions of the CPU and GPU code was revealed. Overall, this comparison of the E90 simulations shows good agreement between the GPU and CPU simulations.

Figures 7a,b show the July 2017 simulation results for the stratospheric synthetic tracer ST80. Figures 7e,f show the simulation results for the northern hemisphere tracer NH50. Similar to E90, the comparisons for ST80 and NH50 show that individual differences between the monthly mean zonal means from the CPU and GPU simulations are mostly below $\pm 5$ % of the maximum volume mixing ratios and that the largest differences are found in regions where sharp gradients are present in the tracer fields. Recognizing the effects of the spin-up time of the simulations and that stratospheric transport has much longer timescales than tropospheric transport, which causes some latency in particular for ST80, similar results are also found for the other months of the year 2017.





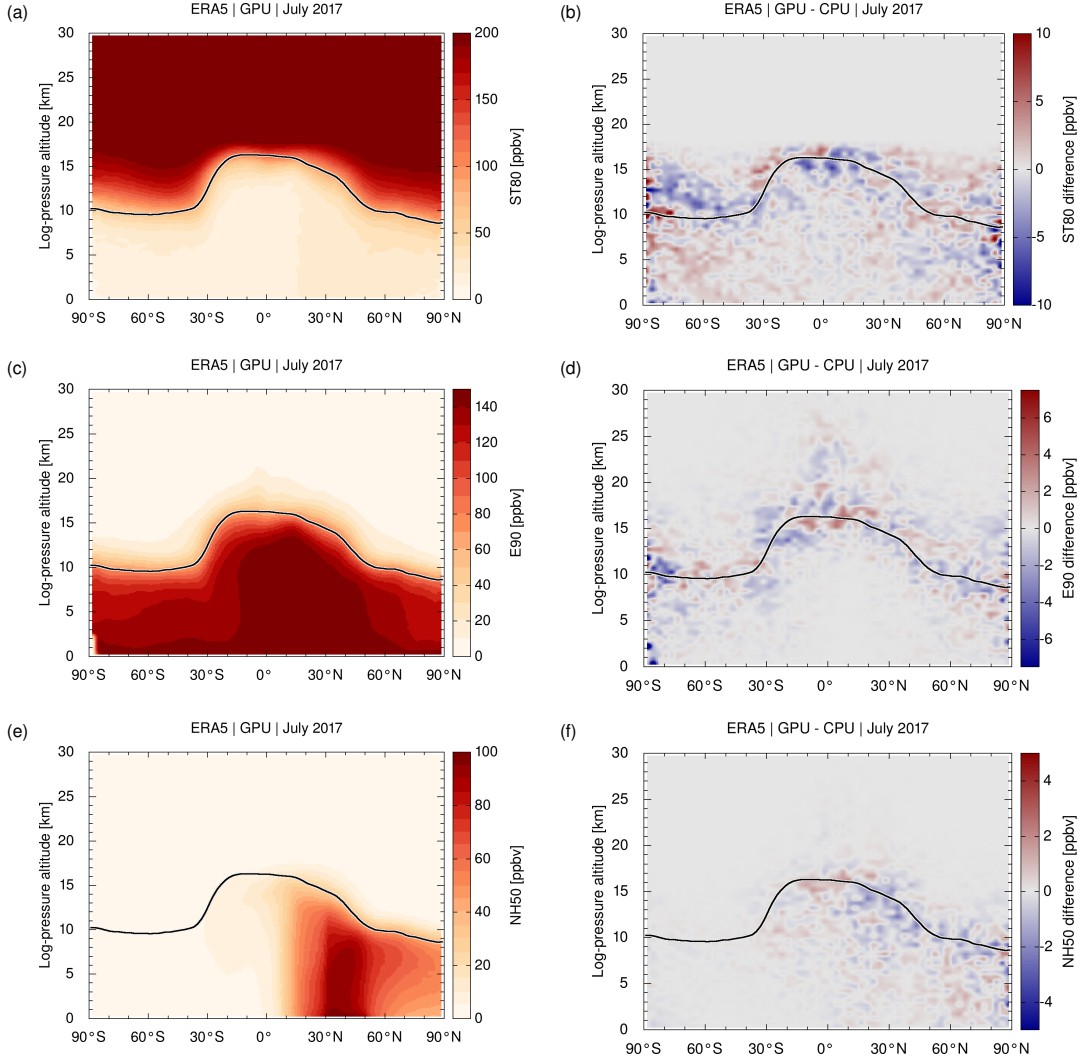

**Figure 7.** Monthly mean zonal means of the synthetic tracers ST80, E90, and NH50 in July 2017 from GPU calculations (left column) as well as corresponding differences between GPU and CPU simulations (right column). Simulations were initialized on 1 January 2017, 00:00 with $10^6$ homogeneously distributed trajectory seeds in the troposphere and stratosphere and are based on ERA5 meteorological input data. The synthetic tracer data are sampled on $3°$ latitude and $500\,\text{m}$ altitude bins. The black curve shows the monthly mean zonal mean dynamical tropopause.

## 3.4 OpenMP scaling analysis

Some parts of the MPTRAC code cannot or have not been ported to GPUs. This includes the file input and output operations, 755 which are directed via the hosting CPUs and the CPU memory, and the functions that are used for preprocessing of the meteorological data, such as the codes to calculate geopotential heights, potential vorticity, convective available potential





energy, the planetary boundary layer, the tropopause height, and others. For those parts of the MPTRAC code that have not been ported to GPUs, it is important to study the OpenMP scalability on the host devices. In particular, good OpenMP scaling up to 12 threads on the JUWELS Booster needs to be demonstrated, as this is the fraction of physical cores shared by each
GPU device on a JUWELS compute node.

Here, we focus on an OpenMP strong scaling test conducted by means of binaries compiled with the PGI compiler, as this is the recommended compiler for GPU applications on JUWELS Booster. For this test, the pinning of the threads to the CPU cores was optimized compared to the default settings of the system by using a fixed assignment of the threads to the sockets of the nodes (by setting the environment variables `OMP_PROC_BIND = true` and `OMP_PLACES = sockets`,
respectively). We measured the runtime and speedup for different numbers of OpenMP threads for a fixed set of $10^6$ globally distributed particles for 24 h of simulation time starting 1 January 2017, 00:00 UTC. The trajectory seeds for this test are the same as those used for comparing the kinematic trajectory calculations in Sect. 3.2. However, similar to the synthetic tracer simulations described in Sect. 3.3, we did not only calculate kinematic trajectories, but also included the effects of turbulent diffusion and subgrid-scale wind fluctuations as well as convection in these calculations.

The results of the OpenMP strong scaling test are shown in Fig. 8. The runtimes indicate that the calculations in the physics modules (trajectories, diffusion, convection, etc.) and the meteorological data processing yield the major contributions to total runtime up to 12 OpenMP threads. At larger numbers of threads, the file input becomes the leading contribution to the total runtime. Concerning the GPU simulations, the OpenMP scaling of the meteorological data processing on the CPUs is of interest, as this is the leading component in the calculations that was not ported to GPUs. In contrast, the scaling of the physics
calculations can be ignored here for now, as these calculations were ported to GPUs. For the meteorological data processing, OpenMP provides a speedup of 10.7 (a parallel efficiency of 90 %) up to 12 threads and 20.2 (84 %) up to 24 threads (Fig. 8b). If the simulations cover more than one socket (48 threads), the parallel efficiency drops to 65 %. Enabling simultaneous multithreading (SMT) provides a further improvement of the speedup from 31.3 at 48 threads to 37.2 at 96 threads. The OpenMP scaling of the meteorological data processing is particularly efficient compared to other parts of the code. This might
be related to the fact that we selected a rather suitable memory layout for the meteorological data. Vertical columns of the meteorological data are ordered sequentially in memory, which allows for efficient use of memory caches as the preprocessing of the meteorological data is mostly conducted on vertical columns of the data.

### 3.5 GPU scaling analysis

In this section, we discuss the GPU scaling of MPTRAC simulations conducted on JUWELS Booster nodes with respect to the
problem size, i.e., the number of particles or trajectories that are calculated. The analysis of the scaling behavior with respect to the problem size is of interest because the NVIDIA A100 GPUs provide a particularly large number of parallel compute elements (Sect. 3.1), which require a high degree of parallelism of the problem in order fully exploit the GPU computing capacities. Here, we tested problem sizes of $10^0$ to $10^8$ particles in the simulations. We measured the runtime for simulations starting on 1 January 2017, 00:00 UTC, covering 24 h of simulation time. The simulations were driven by hourly ERA5 input
data. Particle and grid output were written every 6 h.



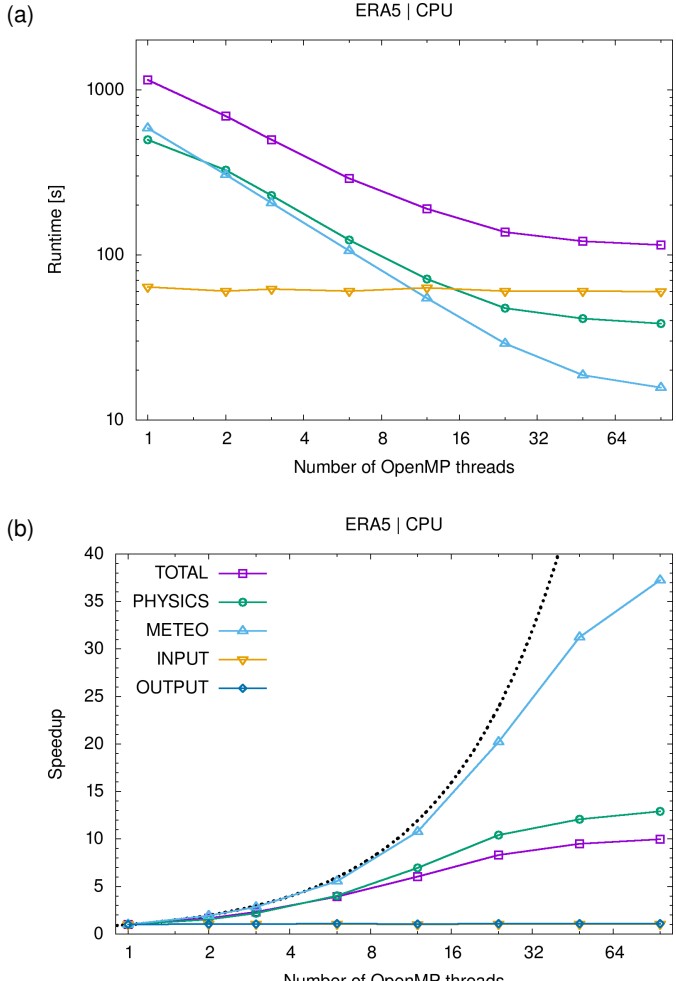

**Figure 8.** Runtime (a) and speedup (b) of an OpenMP strong scaling test for MPTRAC simulations on JUWELS Booster CPUs. The simulations cover a 24 h time period starting on 1 January 2017, 00:00 UTC and use ERA5 input data. The simulations were initialized with a fixed number of $10^6$ particles with homogeneously distributed trajectory seeds. Colored curves show the OpenMP scaling for different parts of the code (see text for details). The black dotted curve indicates linear speedup.

Figure 9a shows the runtime of the MPTRAC simulations required for different numbers of particles on the JUWELS Booster A100 GPU devices. The scaling analysis shows that the total runtime required for the simulations is rather constant and in the range of 122 s to 127 s between $10^0$ to $10^6$ particles. For larger numbers of particles, linear scaling is observed. The total runtime increases up to 153 s for $10^7$ particles and 409 s for $10^8$ particles.

The analysis of individual contributions shows that reading of the ERA5 input data (60 to 64 s, referred to as INPUT in Fig. 9a) and preprocessing of the meteorological data (60 to 62 s, referred to as METEO) accounts for most of the total runtime for $\leq 10^7$ particles. The runtime for this part of the code depends on the size of the meteorological input data and is independent

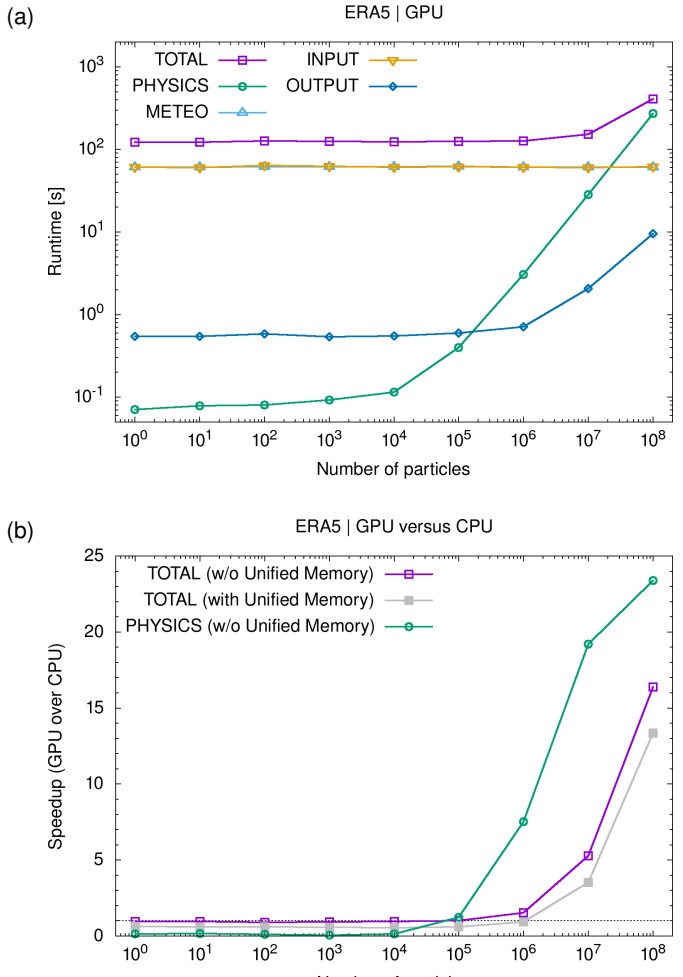

**Figure 9.** Scaling analysis showing (a) the runtime of MPTRAC simulations with respect to the number of particles on a JUWELS Booster GPU node and (b) the speedup of GPU over CPU-only calculations. The simulations cover a 24 h time period starting on 1 January 2017, 00:00 UTC and used ERA5 input data. Colored curves in (a) refer to different parts of the code (see plot key). The speedup of GPUs over CPUs shown in (b) was calculated for simulations with and without using NVIDIA's Unified Memory capability.

of the number of particles. The computing time required to calculate the trajectories, diffusion, convection, and the decay of particle mass (referred to as PHYSICS) is rather constant (0.1 to 0.4 s) for $\leq 10^4$ particles, followed by linear scaling for larger numbers of particles. A transition from constant to linear scaling in the range of $10^3$ to $10^4$ particles can be expected, as at least 3456 particles must be simulated at once, in order to fully exploit the A100 parallel compute capacities (Sect. 3.1). For smaller numbers of particles, a fraction of the compute elements of the GPU device remains unused. Beyond $3 \times 10^7$ particles, the runtime required for the physics calculations exceeds the runtime required for file input and meteorological data preprocessing. The runtime for file output can be neglected in the current setup.






The same scaling test was used to estimate the speed-up achieved by applying the GPU device over a simulation that was conducted on CPUs only. Note that defining a GPU-over-CPU speedup is a difficult task, in general, as the results will depend not only on the GPU device, but on the individual compute capacities of both, the CPU and GPU devices. Nevertheless, to estimate this speed-up, we conducted corresponding CPU-only simulations and runtime measurements by applying 12 cores of a JUWELS Booster compute node. A number of 12 cores was chosen here, as this corresponds to the number of physical

cores sharing a GPU device on a booster node. We think that this specific approach of estimating the GPU-over-CPU speedup provides a fair and more realistic comparison in contrast to comparing the GPU compute capacity to just a single CPU core, for example.

    The estimated speed-up due to applying the GPU devices of a JUWELS Booster compute node is shown in Fig. 9b. In case of not using the "managed memory" option (also referred to as NVIDIA Unified Memory), which corresponds to the results

shown earlier, we found a GPU-over-CPU speedup of less than one for $\leq 10^5$ particles, i. e., the GPU simulations are actually slower than the CPU-only simulations. This behavior changes starting from about $10^6$ particles, with an actual speedup of about 1.5 for $10^6$ particles and about 5.3 for $10^7$ particles. The GPU-over-CPU speedup finally saturates near 16.4 for $10^8$ particles. At these particle numbers, the speedup of the total runtime is largely determined by the scaling for the physics calculations. Additionally, we tested NVIDIA's Unified Memory capability, which allows the programmer to avoid specifying explicit data

transfers between the host and the device, by triggering those data transfers automatically, if memory cache misses occur. It is convenient not having to specify the data transfers explicitly in the code, but our tests showed that the managed memory option causes a performance degradation of about 20 % in the GPU-over-CPU speedup.

    Finally, we conducted a more detailed runtime profile analysis of the GPU simulations based on about 40 individual timers, which are implemented directly into the code. Table 5 shows the results of the profile analysis for the largest GPU simulation

conducted here, covering $10^8$ particles. This is the most compute-intensive simulation, in which about 67 % of the computing time are spent in the physics modules of MPTRAC. The calculation of the trajectories (referred to as MODULE_ADVECTION in Tab. 5) require about 47 % of the runtime. Adding stochastic perturbations to the trajectories to represent diffusion and subgrid-scale wind fluctuations requires 3 % and 7 % of the runtime, respectively. Simulating convection and enforcing particles to remain within the model domain both require about 3 – 4 % of the runtime.





**Table 5.** Runtime profile of an MPTRAC simulation using ERA5 input data with $10^8$ particles for 24 h simulation time. Runtime measurements were conducted on a JUWELS Booster GPU node. Timers with less than 0.01 s are not shown.

| Timer name | Runtime | |
|---|---|---|
| | [s] | [%] |
| TIMER_TOTAL | 409.00 | 100.00 |
| TIMER_INPUT | 62.19 | 15.20 |
| TIMER_GET_MET | 0.64 | 0.16 |
| TIMER_READ_ATM | 1.36 | 0.33 |
| TIMER_READ_MET_LEVELS | 59.77 | 14.61 |
| TIMER_READ_MET_SURFACE | 0.42 | 0.10 |
| TIMER_METPROC | 60.04 | 14.68 |
| TIMER_READ_MET_CAPE | 22.09 | 5.40 |
| TIMER_READ_MET_CLOUD | 1.99 | 0.49 |
| TIMER_READ_MET_EXTRAPOLATE | 3.51 | 0.86 |
| TIMER_READ_MET_GEOPOT | 20.12 | 4.92 |
| TIMER_READ_MET_PBL | 0.95 | 0.23 |
| TIMER_READ_MET_PV | 2.58 | 0.63 |
| TIMER_READ_MET_TROPO | 8.80 | 2.15 |
| TIMER_PHYSICS | 272.70 | 66.68 |
| TIMER_MODULE_ADVECTION | 191.84 | 46.90 |
| TIMER_MODULE_CONVECTION | 15.17 | 3.71 |
| TIMER_MODULE_DECAY | 3.31 | 0.81 |
| TIMER_MODULE_METEO | 6.26 | 1.53 |
| TIMER_MODULE_POSITION | 13.04 | 3.19 |
| TIMER_MODULE_TURBDIFF | 12.68 | 3.10 |
| TIMER_MODULE_TURBMESO | 29.68 | 7.26 |
| TIMER_TIMESTEPS | 0.72 | 0.18 |
| TIMER_OUTPUT | 9.58 | 2.34 |
| TIMER_WRITE_ATM | 5.19 | 1.27 |
| TIMER_WRITE_GRID | 4.39 | 1.07 |
| TIMER_MEMORY | 4.47 | 1.10 |
| TIMER_ACC_INIT | 0.29 | 0.07 |
| TIMER_CREATE_DATA_REGION | 0.02 | 0.00 |
| TIMER_FREE | 0.02 | 0.01 |
| TIMER_UPDATE_DEVICE | 0.85 | 0.21 |
| TIMER_UPDATE_HOST | 3.29 | 0.81 |





Next to the physics modules, other parts of the code required about 33 % if the overall runtime of this large-scale simulation. Reading the input data required about 15 % of the runtime, where most time was needed to read the 3-D level data of the meteorological input data. A similar amount of time (about 15 %) was required for the processing of the meteorological data, with most time spent on calculating CAPE (5.4 %), geopotential heights (4.9 %), and the tropopause (2.1 %). The runtime required for output was 2.4 % in this simulation. However, it needs to be considered that output was written only 6-hourly,

here. The runtime required for output would scale accordingly for more frequent output. The runtime required for data transfers between GPU and CPU memory was also about 1 %, with most time spent on transferring the input meteorological data from CPU to GPU memory. Although this profile analysis did not reveal any major bottlenecks, there is room for further improving the code of MPTRAC. Most attention should be devoted to further optimizing the advection module, as it requires most of the runtime for large-scale simulations.

## 3.6 Timeline analysis of GPU simulations

For a more detailed analysis of the large-scale MPTRAC simulation comprising $10^8$ particles as discussed in Sect. 3.5, we analyzed the timeline of the simulation generated with the NVIDIA Nsight Systems performance analysis tool. Figure 10a shows the timeline for the entire 24 h simulation time period, completed in a runtime of 460 s. Note that the runtime of the Nsight Systems run comprises an overhead of about 50 s compared to the corresponding runtime profile presented in Sect. 3.5.

Figure 10b shows a zoom of the timeline for about 18 s of runtime, which comprises the simulation time period in between two time steps of the ERA5 data.

The black bars on top of Figs. 10a and b show the CPU utilization of a selected subset of the 96 virtual threads operating on a JUWELS compute node. The analysis of the CPU utilization shows that a single CPU, the master thread, is utilized during most of the runtime, i. e, to conduct computation, file-I/O, or data transfers with the GPU, whereas the remaining 11 threads

assigned to a GPU device are utilized via OpenMP only when the meteorological data preprocessing is conducted.

The blue bars in the middle of Figs. 10a and b indicate the GPU utilization over time. In agreement with Sect. 3.5, the timeline analysis shows regular patterns over time (Fig. 9a), where in each block about 30 % of the runtime was spent on file-I/O and meteorological data preprocessing whereas about 70 % of the runtime was used for the physics calculations on the GPUs. On the GPU devices, the timeline analysis shows that 97.3 % of the time was spent in the compute kernels whereas 2.7 %

of the time was required for memory transfers. For the data transfers, most of the time (83.4 %) was spent on host-to-device transfers of the ERA5 data.

The colored bars at the bottom of Figs. 9a and b provide information for the NVIDIA Tools Extension (NVTX) markers, which were inserted directly into the MPTRAC model to indicate distinct sections of the code for profiling. The bars near the bottom also provide access to profiling information on the operating system (OS) runtime libraries (which are utilized for

file-I/O, for instance) as well as the OpenACC und CUDA application programming interfaces.

Overall, the timeline analysis indicates that the code is providing a high utilization rate of the GPU devices, which is a basic requirement for using the GPU devices efficiently. However, the timeline analysis also reveals optimization opportunities. Overlapping file-I/O and GPU computations can hide the file-I/O costs with the expense of a slight increase in memory usage





**Figure 10.** Timeline analysis of a 24 h MPTRAC simulation comprising $10^8$ particles driven by the ERA5 reanalysis for 1 January 2017. Screenshots from the NVIDIA Nsight Systems performance analysis tool show the timeline for (a) the entire simulation and (b) a zoom for 1 h simulation time (in between two time steps of the ERA5 input data). See text for details.

and reduce the wall clock time significantly. As the meteorological data are finally required on the GPU devices to conduct the
Lagrangian transport simulations, it could also be beneficial to port the meteorological data processing from the CPU to the GPU devices to accelerate the processing.





Finally, a more detailed inspection of the physics calculations on the GPUs shows that about 70 % of the time is spent in the advection module, which is used to solve the trajectory equation. A more detailed analysis indicates that this part of the code is memory-bound, i. e., the runtime is limited by the GPU's global memory bandwidth. Future work should focus on optimizing this specific bottleneck of the GPU code by improving data locality and memory access patterns on the GPU devices.

### 3.7 Multi-GPU usage and MPI scaling test

MPTRAC provides a hybrid MPI/OpenMP/OpenACC parallelization to fully exploit the compute capacities of the JUWELS Booster. Each compute node of the booster is equipped with four NVIDIA A100 Tensor Core GPU devices. In order to utilize the capacities of the compute nodes, a multi-GPU approach is required. In MPTRAC, multiple GPU devices can be utilized by means of the MPI parallelization. With the current approach, each MPI task is assigned a single GPU device to offload computations, i. e., four MPI tasks will be operating on each compute node in the multi-GPU mode. At the same time, each MPI task can access up to 12 physical CPU cores of each node by means of the OpenMP parallelization. This scheme is particularly suited for ensemble simulations, i. e., to run many distinct MPTRAC simulations as parallel MPI tasks.

Liu et al. (2020) demonstrated good strong and excellent weak scaling of the MPI/OpenMP hybrid parallelization of MPTRAC in an inverse modeling approach for up to 38400 computing processes on the Tianhe-2 supercomputer at the National Supercomputer Center in Guangzhou, China. As the MPI approach of MPTRAC is embarrassingly parallel and does not require any communication between the MPI tasks, performance degradation is generally not expected to be related to computation or communication, but might arise from other limiting factors, such as file-I/O.

Figure 11 shows the results of an MPI weak scaling test on the JUWELS Booster, utilizing the MPI/OpenMP/OpenACC hybrid parallelization. Here, the test setup was slightly modified compared to earlier work. We conducted the simulations with $5 \times 10^7$ particles. This number of particles was chosen to achieve similar fractions of runtime required for file-I/O, meteorological data preprocessing, and the physics calculations. File-I/O was given a significant fraction of the total runtime in this experiment (at least 25 %), as we expected this part to be most sensitive to performance degradation in the MPI scaling test. The simulation time was reduced from 24 h to 6 h, to limit the overall consumption of computing resources. During the 6 h simulation time, ERA5 data for seven synoptic time steps were read from the global file system. Particle data and grid output were written once at the end of the 6 h time interval.

We measured the total runtime as well as the runtime for the physics calculations, meteorological data processing, file input, and file output for each MPI task. Figure 11 shows the means and standard deviations of the runtime between 1 and 256 MPI tasks. The scaling test shows that the runtime for the meteorological data preprocessing and physics calculations is rather constant over the complete range of MPI tasks tested here. The runtime for file-I/O is nearly constant up to 64 MPI tasks, whereas it significantly increases and shows larger fluctuations for 128 and 256 MPI tasks. Such limitations and fluctuations in runtime for file-I/O are expected because the global file system of the JUWELS Booster is a shared resource and increasing data volumes from our own simulations as well as processes from other users of the system might have affected the runs while the scaling tests were conducted. The MPI scaling test indicates reasonable scaling up to 256 MPI tasks, but it also shows that



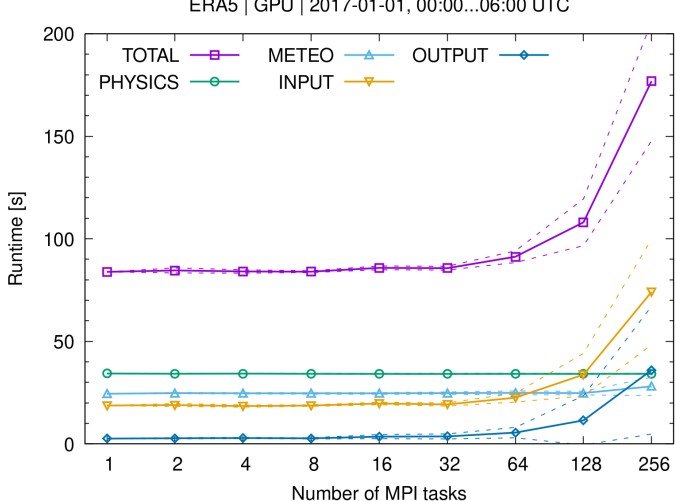

**Figure 11.** MPI weak scaling test for MPTRAC simulations on the JUWELS Booster compute nodes. Each task utilized one GPU device and shared 12 CPU cores. The simulations cover a 6 h time period starting on 1 January 2017, 00:00 UTC and used ERA5 input data. The simulations were initialized with a fixed number of $5 \times 10^7$ particles with homogeneously distributed trajectory seeds. Colored curves show the means (solid) and standard deviations (dashed) of the runtime for different parts of the code.

file-I/O will become a bottleneck for larger simulations, at least if large-scale data sets, such as the ERA5 reanalysis, are used to drive the simulations.

## 4 Conclusions

In this paper, we provide a comprehensive description of the Lagrangian transport model MPTRAC version 2.2. We give an overview on the main features of the model and briefly introduce the code structure. Requirements on the model input data, i. e.,
global meteorological reanalysis or forecast data as well as particle data, are discussed. MPTRAC provides the functionality to calculate various additional meteorological data from basic prognostic variables, such as the 3-D fields of pressure, temperature, winds, water vapor, and cloud ice and liquid water content. This includes functions to calculate geopotential heights and potential vorticity, to calculate additional cloud properties, such as the cloud top pressure and the total column cloud water, to estimate the convective available potential energy, and to determine the boundary layer and the tropopause height level. Some
evaluation of the results of the meteorological data processing code of MPTRAC with data provided along with ECMWF's ERA5 reanalysis is presented.

As its main component, MPTRAC provides an advection module to calculate the trajectories of air parcels based on the explicit mid-point method using given wind fields. Individual stochastic perturbations can be added to the trajectories to account for the effects of diffusion and subgrid-scale wind fluctuations. Additional modules are implemented to simulate the effects
of unresolved convection, wet and dry deposition, sedimentation, hydroxyl chemistry, and other types of exponential loss





processes of particle mass. MPTRAC provides a variety of output options, including the particle data itself, gridded data, profile data, station data, sampled data, ensemble data, and verification statistics and other measures of the model skills. Additional tools are provided to further analyze the particle output, including tools to calculate statistics of particle positions and transport deviations over time.

Next to providing a detailed model description, the focus of this study was to assess the potential for accelerating Lagrangian transport simulations by exploiting graphics processing units (GPUs). We ported the Lagrangian transport model MPTRAC to GPUs by means of the OpenACC programming model. The GPU porting mainly comprised (i) creating a large data region over most of the code to keep the particle data in GPU memory during the entire simulation, (ii) implementing data transfers of the meteorological input data from CPU to GPU memory, (iii) implementing data transfers of the particle data from GPU to CPU

memory for output, (iv) parallelization and offloading of the compute loops for the trajectories and other physical processes of the particles, and (v) removing calls to the GNU Scientific Library, which is not available for GPUs, and adding calls to the cuRAND library for random number generation on the GPUs. Next to various minor changes and fixes, about 60 OpenACC pragmas had to be introduced into the code to manage the data transfers and to offload the compute loops. With the OpenACC approach used here, it was possible to maintain the same code base for both, the CPU and the GPU version of MPTRAC.

We verified and evaluated the GPU version of MPTRAC on the JUWELS Booster at the Jülich Supercomputing Centre, providing access to the latest generation of NVIDIA A100 Tensor Core GPUs. The verification is mostly focusing on comparison of CPU- and GPU-based simulations. A direct comparison of kinematic trajectories showed negligible deviations between the CPU and GPU code for up to 30 days of simulation time. The relative deviations slowly increase to about 0.3 % in the horizontal domain and 0.4 % in the vertical domain after 60 days. To evaluate the impact of additional processes, i. e. the diffusion and

subgrid-scale wind fluctuations, convection, and exponential loss of particle mass, we conducted CPU and GPU simulations of synthetic tracers. The simulations for the tropospheric tracers E90 and NH50 as well as the stratospheric tracer ST80 showed that the differences between the monthly mean zonal means from the CPU and GPU simulations are well below $\pm 10$ % of the maximum volume mixing ratios of the tracers. The largest differences are found in regions where sharp gradients are present in the tracer fields. Overall, these tests are showing very good agreement of the CPU and GPU simulations, i. e., the GPU code

is considered to be verified with respect to the CPU version.

The performance and scaling behavior of the MPI/OpenMP/OpenACC hybrid parallelization of MPTRAC was also assessed on the JUWELS Booster. The OpenMP strong scaling analysis showed satisfying scalability of those parts of the code, which were not ported to GPUs. In particular, the code for meteorological data preprocessing showed an OpenMP parallel efficiency of about 85 to 90 % for up to 12 physical CPU cores of a compute node. The GPU scaling analysis showed that positive speedup

in the GPU simulations compared to CPU-only simulations is achieved for $10^5$ particles, or more. A maximum acceleration due to the utilization of the GPUs of about $16\times$ was achieved for $10^8$ particles. The runtime analysis of the large-scale simulation showed that GPU devices are utilized up to $\sim 70$ % of the time, with most of the time spent in the advection module used to calculate the trajectories. The MPI scaling test showed that MPTRAC can be used to conduct multi-GPU simulations and file-I/O was identified as a potential bottleneck for simulations based on ERA5 comprising more than 256 MPI tasks.

We identified several opportunities for potential improvements of the GPU code of the MPTRAC model in future work. For large-scale simulations, comprising $10^7$ to $10^8$ particles, the trajectory code was found to use about 40 % of the total runtime of the simulations. The efficiency of this part of the code might be improved by optimizing the access to GPU memory and better utilization of the memory caches. However, this optimization would require significant changes of the code of the model, in particular of the data structures and layout. For simulations with fewer particles, the runtime requirements for file-I/O become

more relevant. We started to implement asynchronous file-I/O for a future release of MPTRAC, which allows conducting file-I/O operations on the CPUs at the same time as trajectory calculations are conducted on the GPUs. Nevertheless, we found that GPUs bear large potential to conduct fast and effective Lagrangian transport simulation. The MPTRAC model is considered ready for production runs on the JUWELS booster and other machines applying recent NVIDIA GPUs in its present form.

*Code availability.* MPTRAC is made available under the terms and conditions of the GNU General Public License (GPL) version 3. The

release version 2.2 of MPTRAC described in this paper has been archived on Zenodo (Hoffmann et al., 2021). New versions of MPTRAC are made available via the repository at https://github.com/slcs-jsc/mptrac (last access: 29 November 2021).

*Author contributions.* LH developed the concept for this study and conducted most of the model development, GPU porting, verification and performance analyses. PB and KHM provided expertise on GPU usage and contributed to the performance analysis and optimization of the code. JC, ZC, SG, YH, ML, XW, and LZ are developers and users of the MPTRAC model and provided many suggestions for improving

the model. NT and BV provided expertise on Lagrangian transport modeling and model development. GG and OS were responsible for downloading and preparing the ERA5 data for this study. LH wrote the manuscript with contributions from all co-authors.

*Competing interests.* The authors declare that no competing interests are present.

*Acknowledgements.* The work described in this paper was supported by the Helmholtz Association of German Research Centres (HGF) through the project Pilot Lab Exascale Earth System Modelling (PL-ExaESM). We acknowledge the Jülich Supercomputing Centre for

providing computing time and storage resources on the supercomputer JUWELS and for selecting MPTRAC to participate in the JUWELS Booster Early Access Program. We also acknowledge mentoring and support provided during the 2019 Helmholtz GPU Hackathon. We are thankful to Sebastian Keller, CSCS, who helped us to develop the first version of MPTRAC that was capable of running on GPUs. Xue Wu was supported by the National Natural Science Foundation of China under Grant No. 41975049 and Grant No. 41861134034.



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
