# Peer review of "Massive-Parallel Trajectory Calculations version 2.2 (MPTRAC-2.2): Lagrangian transport simulations on Graphics Processing Units (GPUs)"

_Geoscientific Model Development, 2021_

## Author Comment (AC1)

**Reply to review comments**

We thank the reviewers and the editor for the time and effort spent on the manuscript and for providing helpful comments. We considered all comments and hope that the revised draft properly addresses the open issues. Please find our point-by-point replies below (colored in blue). A revised manuscript with tracked changes has been uploaded.

**Reviewer #1**

The manuscript contains the matter of two articles. The first is a thorough description of the new version of the MPTRAC trajectory code and the second one is the description of the parallelization of this code using GPU which is an excellent example of the application of modern programming methods that is more general than MPTRAC. I accept the choice of the author to group these two works but it would have made sense to make two separate articles to reach a wider audience, at least for the second one.

We agree the manuscript became rather long as both, the model description and the GPU porting of the code are being discussed. While the main focus was mostly on GPU porting in the beginning, we realized it would be helpful to present a detailed and comprehensive model description in a single manuscript, as various individual aspects of the model have been discussed only separately over multiple papers during the last years.

The manuscript contains very useful material, in particular in the second part where it demonstrates how a complex simulation code can be moved to a GPU system using the high level library OpenACC with relatively small effort (compared to the full rewriting required by direct use of CUDA low level libraries). This is an important and inspiring contribution.

Initially, the porting of MPTRAC to GPUs appeared as a big challenge to us, but we were surprised and pleased with the effectiveness of using OpenACC for this task. It is nice to note this message became clear from the paper.

I only have a few comments to be accounted in the revised version

In regard of the sophistication of the rest of the code, the treatment of the vicinity of the pole appears very crude and inaccurate. There has been possibly few concern in the applications of MPTRAC so far but this is a point that should be corrected in the next version.

We agree the treatment of the singularities at the poles in MPTRAC is rather simple. However, as pointed out in the paper, the method was carefully tested within a bachelor thesis (Rößler, 2015). Also, an earlier evaluation comparing global trajectory calculations of the CLaMS and MPTRAC models did not reveal any particular problems near the poles (Hoffmann et al., 2019). We recently conducted another test of the trajectory code,

applying idealized wind fields as described by Williamson et al. (1992). In this test, a particle distribution shaped as a cosine-bell is propagating over a time period of 12 days from the Equator over the North Pole and South Pole back towards the initial position. Visual inspection of the initial and final particle positions did not reveal any significant differences (Fig. 1 in this reply). Despite our current method being simple, we think it is sufficient for many applications. Possibly, the methods works with reasonable accuracy, even close to the poles, because all coordinate transformations are calculated with double precision (64 Bits). Nevertheless, we will revisit the method in future work, to see if it can be improved.

The convective parameterization is based on the assumption of CAPE relaxation and an important parameter is the CAPE threshold that should deserve some discussion. The manuscript says that a global value is used but CAPE accumulates much more over the continents than over the ocean, leading to much more intense storms in continental regions. Therefore a single threshold value will probably produce excessive mixing over the continents and too small mixing over the oceans. More generally, it is recognized by all experts in convective parameterization that CAPE alone is a bad predictor of convective onset. As the ERA5 archive the upward and downward convective fluxes resulting from its state of the art parameterization of convection, why not using these data instead of a very crude representation of convection. Again, this might be considered in the next version.

We added the following paragraph in Sect. 2.3.4, to provide a more detailed discussion of the benefits and limitations of the extreme convection parametrization and references to related work: "The extreme convection parametrization implemented here is arguably a rather simple approach, as it relies only on CAPE and the equilibrium level from the meteorological input data. Nevertheless, first tests showed that this parametrization significantly improves transport patterns in the free troposphere. We also conducted experiments to further improve the parametrization by considering additional parameters such as the convective inhibition (CIN) to improve the onset of convection. More sophisticated convection parametrizations have been developed and implemented in Lagrangian transport models during recent years (Forster et al., 2007; Brinkop and Jöckel, 2019; Konopka et al., 2019; Wohltmann et al., 2019). These parametrizations consider additional meteorological variables such as convective mass fluxes and detrainment rates that originate from external convective parameterizations applied in state-of-the-art forecasting models. Future work may look at evaluating the extreme convection parametrization and implementing more advanced convection parametrizations in MPTRAC." We are currently evaluating the extreme convection parameterization used in MPTRAC in a separate study.

The manuscript fails to quote this work "optimization of atmospheric transport models on HPC platforms, de la Cruz et al., Computers & Geosciences, 2016, doi:10.1016/j.cageo.2016.08.019" which addresses very similar issues.

Thank you for pointing this out. We added the reference to the introduction.

[Figure]

Figure 1: Assessment of trajectory calculations using an idealized wind field with advection over the poles as described by Williamson et al. (1992). Maps show particle distributions after (a) 0 h, (b) 48 h, (c) 120 h, (d) 192 h, (e) 216 h, and (f) 288 h.

Figure 10 made from screen copies is not readable, either on print or on the screen.

We have to apologize, the quality of Fig. 10 is limited, because the NVIDIA tools used here are created for interactive browsing and not meant to produce high-quality graphics. Nevertheless, we hope Fig. 10 is useful in its present form, to illustrate what kind of analysis and insights can be achieved by means of these tools. In case a reader is interested in the particular details of the timeline analysis, we uploaded the sampling report (qdrep- and SQL-file to browse with NVIDIA Nsight Systems) to an open data repository (Haghighi Mood and Hoffmann, 2022).

Other minor comments

185: pressure is not the best choice of vertical transport for Lagrangian transport in the stratosphere as well where many models use instead the potential temperature and heating rates instead of pressure tendencies.

Different approaches for the vertical coordinate and vertical transport in MPTRAC are currently being evaluated as part of a PhD thesis. We added: "The next release of MPTRAC will allow the user to choose between pressure and the isentropic-sigma hybrid coordinate ($\zeta$) of Mahowald et al. (2002) as vertical coordinate."

205: I guess the authors meant linear in log pressure.

We implemented linear interpolation in pressure, which is less accurate but faster than logarithmic interpolation. We replaced "in the vertical domain" by "in pressure", to clarify.

770: The results from OpenMP parallelization may vary a lot according the scheduling strategy. This should be mentioned.

We added "For this test, the OpenMP static scheduling strategy was applied.", to clarify.

896: I do not see any fluctuations but a regular increase in Fig. 11.

Figure 11 shows increasing fluctuations of the runtime for increasing numbers of MPI tasks because the standard deviations (dashed curves) are increasing.

**Reviewer #2**

The manuscript by Hoffmann et al. presents and impressive piece of work. I can only congratulate the authors on the development of MPTRAC and its parallelization on GPUs, which is the main topic of the study. The manuscript is well written and structured and the methods and results sections are easy to follow.

We would like to thank the reviewer for the encouraging statement.

I thus only have a few minor comments, suggestions, and corrections that the authors should consider before publication:

The introduction is quite MPTRAC-centric. Since the focus is on code parallelization, it would be good to include references on parallelization approaches in other Lagrangian dispersion models, e.g. Brioude et al. (2013) for FLEXPART-WRF, Jones et al. (2007) for NAME, Pisso et al. (2019) for other versions of FLEXPART (with MPI or OpenMP parallelization and asynchronous I/O in case of FLEXPART-COSMO). There is actually also a GPU version of (parts of) FLEXPART developed many years ago (https://db.cger.nies.go.jp/metex/flexcpp.html), but unfortunately it was never published in peer-reviewed literature to my knowledge.

We agree that earlier work on the topic of GPU porting of LPDMs should be properly referenced in the introduction. We added: "Large-scale and long-term Lagrangian transport simulations for climate studies or inverse modeling applications can become very compute-intensive (Heng et al., 2016; Liu et al., 2020). The benefits of parallel computing to accelerate Lagrangian transport simulations have been assessed in several studies (Jones et al., 2007; Brioude et al., 2013; Pisso et al., 2019). The aim of the present study is to investigate the potential of using GPUs for accelerating Lagrangian transport simulations, a topic first explored by Molnár et al. (2010). Related efforts to solve the advection–diffusion equation for a 3-D time-dependent Eulerian model on GPUs have been reported by de la Cruz et al. (2016). The Centre for Global Environmental Research at the National Institute for Environmental Studies, Japan, published online a Lagrangian particle dispersion model based on parts of the FLEXPART model (Stohl et al., 2005; Pisso et al., 2019), reporting that performance can be improved by more than 20 times by using a multi-core CPU with an NVIDIA GPU (CGER, 2016)."

The introduction should also explain more clearly, what the main areas of application of MPTRAC are. It seems to be designed primarily to study large scale atmospheric transport in the free troposphere and stratosphere but not for transport and mixing in the atmospheric boundary layer (ABL). This is important to mention, because Lagrangian models are increasingly being used for inverse emission estimation, for which e.g. a proper representation of turbulent mixing in the ABL is critical.

We added the following clarification in the introduction: "Based on the given applications, MPTRAC is designed primarily to study large-scale atmospheric transport in the free troposphere and stratosphere. Applications of the model to the planetary boundary layer are limited as MPTRAC lacks more sophisticated parametrizations of turbulence and mixing required for that region."

The manuscript convinced me that MPTRAC is a technically carefully designed, flexible and computationally efficient model. However, I was less convinced that it is also doing a good job in terms of accurately representing atmospheric transport. A key criterion for Lagrangian particle dispersion models, for example, is the well-mixed condition of Thomson (1987): A tracer well-mixed in the atmosphere should not un-mix due to the simulated transport. This is challenging to achieve but is critical for simulating mixing in the ABL or inversely estimating emissions, for example. Simple mixing schemes (e.g. without density correction term) as implemented in the model lead to un-mixing. It would be good to know

the magnitude of un-mixing generated by the model in long simulations (un-mixing likely saturates at some point). This could be studied in a simulation similar to those presented in Section 3.3, but where particles with uniform mass are initialized proportional to air density. Particle densities should ideally remain proportional to air density throughout the simulation.

Please see reply below.

The synthetic tracer simulations presented in Section 3.3. are suitable to study differences between the CPU and GPU versions, but they are not sufficient to demonstrate that transport is generally well represented in the model. A much more challenging diagnostic for stratospheric transport, for example, would be age of air, which is known to be underestimated by many transport models.

Please see reply below.

I thus strongly encourage the authors to focus on such critical aspects in future studies to provide a thorough scientific benchmark for future applications of the model. This is more a comment than a suggestion for modifying the current publication.

We agree the accuracy and representation of various transport processes in MPTRAC should be evaluated in more detail. However, in particular simulations of age-of-air will require simulation periods of 10 years, or more, to properly reflect the timescales of stratospheric transport. We are afraid, this is beyond the scope of the present study, but we will address it in future work, as suggested.

Small points:

Page 7, line 184: What exactly do you mean by "pushed back"? The standard approach in Lagrangian models is that particles are reflected. "Pushing back" likely leads to accumulation of air parcels at the surface or upper boundary of the model.

Currently, particles will slide along the top (bottom) boundary of the model as defined by the uppermost pressure level (surface pressure) until the particles experience a downdraft (updraft). For example, if an advection or diffusion step cause particle pressure to become larger than the surface pressure, the pressure of the particle will be set to match the surface pressure. This is what we refer to as "pushing back" the particle to the surface. This approach follows the HYSPLIT model. The current version of the HYSPLIT model description states that "advection continues along the surface if trajectories intersect the ground". However, we agree that reflection of the particles might be a better choice, so we added: "The next release of the MPTRAC model will provide a new option to alternatively reflect particles at the top and bottom boundaries, in order to avoid accumulation of particles at the boundaries." The corresponding code has been added to the git repository.

Page 9, line 250: Shouldn't it be $|\phi| > \phi_{max}$? Same issue on the next line on page 10.

Yes, this should be latitude, $\phi$, instead of longitude, $\lambda$. Fixed, thank you.

Page 15: Convection is parameterized in an overly simplified way, since e.g. deep convection does not at all lead to uniform vertical mixing. It would be good to mention (and to consider) more advanced approaches such as Forster et al. (2007, https://doi.org/10.1175/JAM2470.1).

We added the following paragraph in Sect. 2.3.4, to provide a more detailed discussion of the benefits and limitations of the extreme convection parametrization and references to related work: "The extreme convection parametrization implemented here is arguably a rather simple approach, as it relies only on CAPE and the equilibrium level from the meteorological input data. Nevertheless, first tests showed that this parametrization significantly improves transport patterns in the free troposphere. We also conducted experiments to further improve the parametrization by considering additional parameters such as the convective inhibition (CIN) to improve the onset of convection. More sophisticated convection parametrizations have been developed and implemented in Lagrangian transport models during recent years (Forster et al., 2007; Brinkop and Jöckel, 2019; Konopka et al., 2019; Wohltmann et al., 2019). These parametrizations consider additional meteorological variables such as convective mass fluxes and detrainment rates that originate from external convective parameterizations applied in state-of-the-art forecasting models. Future work may look at evaluating the extreme convection parametrization and implementing more advanced convection parametrizations in MPTRAC." We are currently evaluating the extreme convection parameterization used in MPTRAC in a separate study.

Page 18: Also dry deposition is described in a highly simplified way. Dry deposition does not only depend on particle or gas properties but also on the state of the atmosphere (in addition to surface properties). Also here it should be mentioned that more advanced approaches for Lagrangian models exist, e.g. Webster and Thomson (2011, https://doi.org/10.1504/IJEP.2011.047322).

We revised the text to point out that dry deposition depends on both, the surface characteristics and the atmospheric conditions. We also added a sentence to point out that the dry and wet deposition parametrizations will have to be compared to more advanced parametrizations (e.g., Webster and Thomson, 2011, 2014).

Page 30: Which number of compute cores of the GPU is the most relevant number for MPTRAC? Is it the number of FP32 or FP64 cores? Later it becomes clear that it is the latter. Is double precision really needed? Did you test MPTRAC with single precision?

To clarify, we rephrased: "Each NVIDIA A100 GPU comprises 6912 INT32 and 6912 FP32 compute cores as well as 3456 FP64 compute cores, the latter being most relevant for us because most calculations in MPTRAC are conducted at double precision." Although this is an interesting topic, we did not systematically assess differences between single precision and double precision arithmetic. At present, we would not expect this to cause large differences as the code is memory-bound and not compute-bound.

Figure 7: The differences between GPU and CPU simulations presented in panels b), d) and f) are likely due to statistical noise. This could be shown by performing multiple CPU

simulations with different random seeds and evaluate the differences in the same way as the differences between CPU and GPU.

We repeated the Lagrangian transport simulation for the artificial tracers shown in Fig. 7, but we changed the seeds for the CPU simulations, as suggested. For reference, the results are shown in Fig. 2 in this reply. The test indicates that the differences between the CPU and GPU simulations are largely due to statistical noise. We added the following sentence to point this out: "The remaining differences are attributed to statistical noise, as can be seen by running simulations with different random seeds."

Section 3.7: I didn't quite understand this scaling test. Why does the runtime shown in Fig. 11 not decrease with the number of MPI tasks? What is the difference between a weak and a strong scaling test?

To make the text more clear, we added: "Here, strong scaling is defined as how the runtime varies with the number of compute elements for a fixed total problem size. Weak scaling is defined as how the runtime varies with the number of compute elements for a fixed problem size per compute element." Following this, the runtime in the weak scaling test is expected to remain constant with respect to the problem size.

Small corrections and typos:

Page 11, Line 272: Change to "The following choices are made .."

Changed as suggested.

Page 23, Line 500: shouldn't it be "interpreting" rather than "interpolating"?

We rephrased the sentence as "Altogether, these tools provide a great flexibility in exploiting meteorological data in many applications."

Page 30, line 678: "MPTRAC was build" → "MPTRAC was built"

Fixed.

Page 40, line 830: "33% if the overall runtime" → "33% of the overall runtime"

Fixed.

Page 40, line 857: It should be Figs. 10a and b rather than 9a and b.

Fixed.

**Additional Changes**

We revised the acknowledgements of the paper.

[Figure]

Figure 2: Comparison of CPU and GPU simulations of artificial tracers as shown in Fig. 7 of the manuscript, but initialized with different seeds of the random number generator for the CPU code.